# Revisiting the Calibration of Modern Neural Networks

**Matthias Minderer**   **Josip Djolonga**   **Rob Romijnders**   **Frances Hubis**
**Xiaohua Zhai**   **Neil Houlsby**   **Dustin Tran**   **Mario Lucic**
Google Research, Brain Team
{mjlm, lucic}@google.com

## Abstract

Accurate estimation of predictive uncertainty (model calibration) is essential for the safe application of neural networks. Many instances of miscalibration in modern neural networks have been reported, suggesting a trend that newer, more accurate models produce poorly calibrated predictions. Here, we revisit this question for recent state-of-the-art image classification models. We systematically relate model calibration and accuracy, and find that the most recent models, notably those not using convolutions, are among the best calibrated. Trends observed in prior model generations, such as decay of calibration with distribution shift or model size, are less pronounced in recent architectures. We also show that model size and amount of pretraining do not fully explain these differences, suggesting that architecture is a major determinant of calibration properties.

## 1   Introduction

Neural networks, especially vision models, are increasingly used in safety-critical applications such as autonomous driving (Bojarski et al., 2016), medical diagnosis (Esteva et al., 2017; Jiang et al., 2012), and meteorological forecasting (Sønderby et al., 2020). For such applications, it is essential that model predictions are not just accurate, but also well calibrated. Model calibration refers to the accuracy with which the scores provided by the model reflect its predictive uncertainty. For example, in a medical application, we would like to defer images for which the model makes low-confidence predictions to a physician for review (Kompa et al., 2021). Skipping human review due to confident, but incorrect, predictions, could have disastrous consequences.

While intense research and engineering effort has focused on improving the predictive accuracy of models, less attention has been given to model calibration. In fact, over the last few years, there have been many reports that calibration of modern neural networks can be surprisingly poor, despite the advances in accuracy (e.g. Guo et al. 2017; Lakshminarayanan et al. 2017; Malinin & Gales 2018; Thulasidasan et al. 2019; Hendrycks et al. 2020b; Ovadia et al. 2019; Wenzel et al. 2020; Havasi et al. 2021; Rahaman & Thiery 2020; Leathart & Polaczuk 2020). Some works suggest a trend for larger, more accurate models to be worse calibrated (Guo et al., 2017).

These concerns are more relevant than ever, since the architecture size, amount of training data, and computing power used by state-of-the-art models continue to increase. At the same time, rapid advances in model architecture (Tolstikhin et al., 2021; Dosovitskiy et al., 2021) and training approaches (Chen et al., 2020; Mahajan et al., 2018; Radford et al., 2021) raise the question whether past results on calibration, largely obtained on standard convolutional architectures, extend to current state-of-the-art models. Since model advances are quickly translated to real-world, safety-critical applications (e.g. Mustafa et al. 2021), there is an urgent need to re-assess the calibration properties of current state-of-the-art models.

35th Conference on Neural Information Processing Systems (NeurIPS 2021).

**Contributions.** To address this need, we provide a systematic comparison of recent image classification models, relating their accuracy, calibration, and design features. We find that:

1. The best current models, including the non-convolutional MLP-Mixer (Tolstikhin et al., 2021) and Vision Transformers (Dosovitskiy et al., 2021), are well calibrated compared to past models and their performance is more robust to distribution shift.
2. In-distribution calibration slightly deteriorates with increasing model size, but this is outweighed by a simultaneous improvement in accuracy.
3. Under distribution shift, calibration *improves* with model size, reversing the trend seen in-distribution.
4. Accuracy and calibration are correlated under distribution shift, such that optimizing for accuracy may also benefit calibration.
5. Model size, pretraining duration, and pretraining dataset size cannot fully explain differences in calibration properties between model families.

Our results suggest that further improvements in model accuracy will continue to benefit calibration. They also hint at architecture as an important determinant of model calibration. We provide code and a large dataset of calibration measurements, comprising 180 distinct models from 16 families, each evaluated on 79 ImageNet-scale datasets and 28 metric variants.[1]

## 2 Related Work

**Measures of model calibration.** The losses that are commonly used to train classification models, such as cross-entropy and squared error, are proper scoring rules (Gneiting et al., 2007) and are therefore guaranteed to yield perfectly calibrated models at their minimum—in the infinite-data limit. However, in practice, due to model mismatch and overfitting, even losses based on proper scoring rules may result in poor model calibration. Miscalibration is commonly quantified in terms of Expected Calibration Error (ECE; Naeini et al. 2015), which measures the absolute difference between predictive confidence and accuracy. We focus on ECE because it is a widely used and accepted calibration metric. Nevertheless, it is well understood that estimating ECE accurately is difficult because estimators can be strongly biased and many estimator variants exist (Nixon et al., 2019; Roelofs et al., 2020; Vaicenavicius et al., 2019; Gupta et al., 2021). Section 5 discusses these issues and our approaches to mitigate them.

Alternatives to ECE include likelihood measures, Brier score (Brier, 1950), Bayesian methods (Gelman et al., 2013), and conformal prediction (Shafer & Vovk, 2008). Further, model calibration can be represented visually with reliability diagrams (DeGroot & Fienberg, 1983). Figure 8 and Appendix F provide likelihoods, Brier scores, and reliability diagrams for our main analyses.

**Empirical studies of model calibration.** There have been many recent empirical studies on the robustness (accuracy under distribution shift) of image classifiers (Geirhos et al., 2019; Taori et al., 2020; Djolonga et al., 2020; Hendrycks et al., 2020a). Several works have also studied calibration. Most notable is Guo et al. (2017), who found that "modern neural networks, unlike those from a decade ago, are poorly calibrated", that larger networks tend to be calibrated worse, and that "miscalibration worsen[s] even as classification error is reduced." Other works have corroborated some of these findings (e.g., Thulasidasan et al. 2019; Wen et al. 2021). This line of work suggests a trend that larger models are worse calibrated, which would have major implications for research toward bigger models and datasets. We show that for more recent models, this trend is negligible in-distribution and in fact reverses under distribution shift.

Ovadia et al. (2019) empirically study calibration under distribution shift and provide a large comparison of methods for improving calibration. They report that both accuracy and calibration deteriorate with distribution shift. While we observe the same trend, we find that the calibration of some recent model families decays so slowly under distribution shift that the decay in accuracy is likely more relevant in practice (Section 4.3).

Ovadia et al. also find that, *across methods for improving calibration*, improvements on in-distribution data do not necessarily translate to out-of-distribution data. This finding may suggest that there is little correlation between in-distribution and out-of-distribution calibration in general. However, our

---

[1]Available at https://github.com/google-research/robustness_metrics/tree/master/robustness_metrics/projects/revisiting_calibration.

results show that, *across model architectures*, the models with the best in-distribution calibration are also the best-calibrated on a range of out-of-distribution benchmarks. The important implication of this result is that designing models based on in-distribution performance likely also benefits their out-of-distribution performance.

**Improving calibration.** Many strategies have been proposed to improve model calibration such as post-hoc rescaling of predictions (Guo et al., 2017), averaging multiple predictions (Lakshmi-narayanan et al., 2017; Wen et al., 2020), and data augmentation (Thulasidasan et al., 2019; Wen et al., 2021). Here, we focus on the intrinsic calibration properties of state-of-the-art model families, rather than methods to further improve calibration.

As a baseline on top of a model's intrinsic calibration properties, we study temperature scaling (Guo et al., 2017). It is effective in improving calibration and so simple that it can be applied in many cases at minimal additional cost, in contrast to many more sophisticated methods. Temperature scaling re-scales a model's logits by a single parameter, chosen to optimize the model's likelihood on a held-out portion of the training data. This temperature factor changes the model's *confidence*, i.e., whether the model predictions are on average too certain (overconfident), optimally confident, or too uncertain (underconfident). The classification accuracy of the model is not affected by temperature scaling. A large fraction of model miscalibration is typically due to average over- or underconfidence, e.g. due to suboptimal training duration (Guo et al., 2017). By normalizing a model's confidence, temperature scaling not only improves calibration, but also removes a primary confounder that can hide trends in calibration between models (see Section 4.2 and Appendix D). Therefore, we study both unscaled and temperature-scaled predictions in the paper.

# 3   Definitions and Notation

We consider the multi-class classification problem, as analyzed by Bröcker (2009), where we observe a variable $X$ and predict a categorical variable $Y \in \{1, 2, \ldots, k\}$. We model our predictor $f$ as a function that maps every input instance $X$ to a categorical distribution over $k$ labels, represented using a vector $f(X)$ belonging to the $(k-1)$-dimensional simplex $\Delta = \{p \in [0,1]^k \mid \sum_{y=1}^{k} p_y = 1\}$.

Intuitively, a model $f$ is well-calibrated if its output truthfully quantifies the predictive uncertainty. For example, if we take all data points $x$ for which the model predicts $[f(x)]_y = 0.3$, we expect 30% of them to indeed take on the label $y$. Formally, the model $f$ is said to be calibrated if (Bröcker, 2009)

$$\forall p \in \Delta \colon P(Y = y \mid f(X) = p) = p_y. \tag{1}$$

We will focus on a slightly weaker, but more practical condition, called top-label or argmax calibration (Kumar et al., 2019; Guo et al., 2017). This requires that the above holds only for the most likely label, i.e., $\forall p^* \in [0,1]$

$$P(Y \in \arg\max p \mid \max f(X) = p^*) = p^*, \tag{2}$$

where the $\max$ and $\arg\max$ act coordinate-wise.

The most common measure of the degree of miscalibration is the *Expected Calibration Error (ECE)*, which computes the expected disagreement between the two sides of eq. (2)

$$\mathbb{E}\big[|p^* - E[Y \in \arg\max f(X) \mid \max f(X) = p^*|\big]. \tag{3}$$

Unfortunately, eq. (3) cannot be estimated without quantization as it conditions on a null event. Hence, one typically first buckets the predictions into $m$ bins $B_1, \ldots, B_m$ based on their top predicted probability, and then takes the expectation over these buckets. Namely, if we are given a set of $n$ i.i.d. samples $(x_1, y_1), \ldots, (x_n, y_n)$ distributed as $P(X, Y)$, then we assign each $j \in \{1, \ldots, n\}$ to a bucket $B_i$ based on $\max f(x_j)$. Then, we compute in each bucket $B_i$ the confidence$(B_i) = \frac{1}{|B_i|} \sum_{j \in B_i} \max f(x_j)$ and the accuracy$(B_i) = \frac{1}{|B_i|} \sum_{j \in B_i} [\![y_j \in \arg\max f(x_j)]\!]$, where $[\![\cdot]\!]$ is the Iverson bracket. Finally, we construct an estimator by taking the expectation over the bins

$$\widehat{\text{ECE}} = \sum_{i=1}^{m} \frac{|B_i|}{n} \left| \text{accuracy}(B_i) - \text{confidence}(B_i) \right|. \tag{4}$$

In Section 5 we discuss the statistical properties of this estimator, possible pitfalls, and several mitigation strategies.

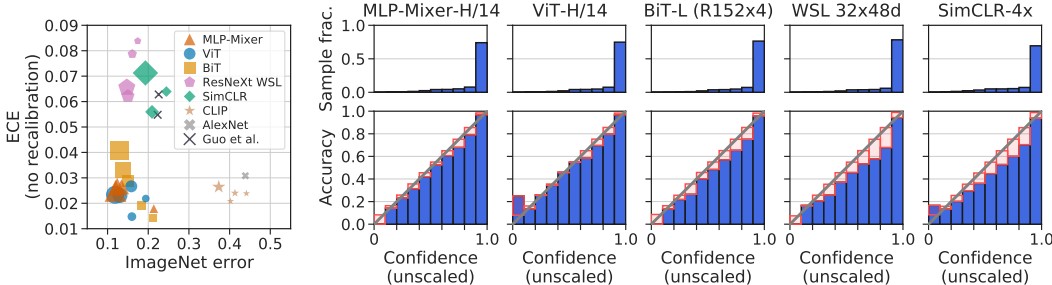

Figure 1: Some modern neural network families are both highly accurate and well-calibrated. Left: Expected calibration error (ECE) vs. classification error on IMAGENET for state-of-the-art image classification models. Marker size indicates relative model size within its family. Points labeled "Guo et al." are the values reported for DenseNet-161 and ResNet-152 in Guo et al. (2017). Right: Confidence distribution (top row) and reliability diagrams (bottom row) for some of the models.

## 4 Empirical Evaluation

### 4.1 Experimental Setup

**Model families.** In this study, we consider a range of recent and some historic state-of-the-art image classification models. Our selection of models covers convolutional and non-convolutional architectures, as well as supervised, weakly supervised, unsupervised and zero-shot training. We follow the original publications in naming the model variants within each family (e.g. different model sizes). See Appendix A.1 for a detailed description of all used models.

1. MLP-Mixer (Tolstikhin et al., 2021) is based exclusively on multi-layer perceptrons (MLPs) and is pre-trained on large supervised datasets.
2. ViT (Dosovitskiy et al., 2021) processes images with a transformer architecture originally designed for language (Vaswani et al., 2017) and is also pre-trained on large supervised datasets.
3. BiT (Kolesnikov et al., 2020) is a ResNet-based architecture (He et al., 2016). It is also pre-trained on large supervised datasets.
4. ResNext-WSL (Mahajan et al., 2018) is based on the ResNeXt architecture and trained with weak supervision from billions of hashtags on social media images.
5. SimCLR (Chen et al., 2020) is a ResNet, pretrained with an unsupervised contrastive loss.
6. CLIP (Radford et al., 2021) is pretrained on raw text and imagery using a contrastive loss.
7. AlexNet (Krizhevsky et al., 2012; Krizhevsky, 2014) was the first convolutional neural network to win the ImageNet challenge.

All models are either trained or fine-tuned on the IMAGENET training set, except for CLIP, which makes zero-shot predictions using IMAGENET class names as queries.

**Datasets.** We evaluate accuracy and calibration on the IMAGENET validation set and the following out-of-distribution benchmarks using the Robustness Metrics library (Djolonga et al., 2020):

1. IMAGENETV2 (Recht et al., 2019) is a new IMAGENET test set collected by closely following the original IMAGENET labeling protocol.
2. IMAGENET-C (Hendrycks & Dietterich, 2019) consists of the images from IMAGENET, modified with synthetic perturbations such as blur, pixelation, and compression artifacts at a range of severities.
3. IMAGENET-R (Hendrycks et al., 2020a) contains artificial renditions of IMAGENET classes such as art, cartoons, drawings, sculptures, and others.
4. IMAGENET-A (Hendrycks et al., 2021) contains images that are classified as belonging to IMAGENET classes by humans, but adversarially selected to be hard to classify for a ResNet50 trained on IMAGENET.

For the post-hoc recalibration of models, we reserve 20% of the IMAGENET validation set (randomly sampled) for fitting the temperature scaling parameter. All reported metrics are computed on the remaining 80% of the data. For evaluations on IMAGENET-C, we also exclude the 20% of images that are based on the IMAGENET images used for temperature scaling.

**Calibration metric.**  Throughout the paper, we estimate ECE using equal-mass binning and 100 bins. Appendix E shows that our results hold for other ECE variants and are consistent with the Brier score and model likelihood.

## 4.2  In-Distribution Calibration

We begin by considering ECE on clean IMAGENET images (referred to as in-distribution). Figure 1 shows in-distribution ECE and reliability diagrams before any recalibration of the predicted probabilities. We find that several recent model families (MLP-Mixer, ViT, and BiT) are both highly accurate *and* well-calibrated compared to prior models, such as AlexNet or the models studied by Guo et al. (2017). This suggests that there may be no continuing trend for highly accurate modern neural networks to be poorly calibrated, as suggested previously (Guo et al., 2017; Lakshminarayanan et al., 2017; Malinin & Gales, 2018; Thulasidasan et al., 2019; Hendrycks et al., 2020b; Ovadia et al., 2019; Wenzel et al., 2020; Havasi et al., 2021; Rahaman & Thiery, 2020; Leathart & Polaczuk, 2020). In addition, we find that a recent zero-shot model, CLIP, is well-calibrated given its accuracy.

**Temperature scaling reveals consistent properties of model families.**  The poor calibration of past models can often be remedied by post-hoc recalibration such as temperature scaling (Guo et al., 2017), which raises the question whether a difference between models remains after recalibration. We find that the most recent architectures are better calibrated than past models *even after temperature scaling* (Figure 2, right).

More generally, temperature scaling reveals consistent trends in the calibration properties between families that are obscured in the unscaled data by simple over- or underconfidence miscalibration. Before temperature scaling (Figure 2, left), several families overlap in their accuracy/calibration properties (MLP-Mixer, ViT, BiT). After temperature scaling (Figure 2, right), a clearer separation of families and consistent trends between accuracy and calibration within each family become apparent. Notably, temperature scaling reconciles our results for BiT (a ResNet architecture) with the results reported by Guo et al. for ResNets trained on IMAGENET.

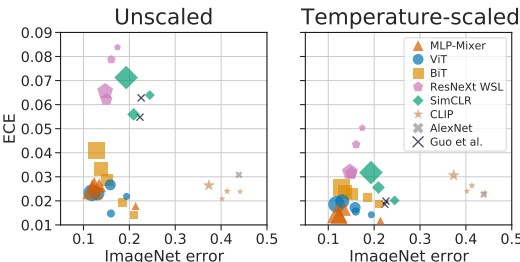

Figure 2: Temperature scaling reveals consistent properties of model families. Left: ECE vs. classification error as in Figure 1. Right: ECE after applying temperature scaling.

Furthermore, models pretrained without additional labels (SimCLR) or with noisy labels (ResNeXt-WSL) tend to be calibrated worse for a given accuracy than ResNets trained with supervision (BiT and the models studied by Guo et al.). Finally, non-convolutional model families like MLP-Mixer and ViT can perform just as well, if not better, than convolutional ones.

**Differences between families are not explained by model size or pretraining amount.**  We next attempt to disentangle how the differences between model families affect their calibration properties. We focus on model size and amount of pretraining, both important trends in state-of-the-art models.

We first consider model size. Prior work has suggested that larger neural networks are worse calibrated (Guo et al., 2017). We also find that within most families, larger members tend to have higher calibration error (Figure 2, right). However, at the same time, larger models have consistently lower classification error. This means that each model family occupies a different Pareto set in the tradeoff between accuracy and calibration. For example, our results suggest that, at any given accuracy, ViT models are better calibrated than BiT models. Changing the size of a BiT model cannot move it into the Pareto set of ViT models. Model size can therefore not fully explain the intrinsic calibration differences between these model families.[2]

We next consider model pretraining. Many current state-of-the-art image models use transfer learning, in which a model is pre-trained on a large dataset and then fine-tuned to the task of interest (Kolesnikov et al., 2020; Chen et al., 2020; Xie et al., 2020). With transfer learning, large data sources can be

---

[2]This relationship between model size, accuracy and calibration holds for all families we study except ResNeXt-WSL, for which increasing model sizes improves *both* accuracy *and* calibration. While investigating this difference was out of the scope of this work, it may be a promising direction for future research.

exploited to train the model, even if little data are available for the final task. To test how the amount of pretraining affects calibration, we compare BiT models pretrained on IMAGENET (1.3M images), IMAGENET-21K (12.8M images), or JFT-300 (300M images; Sun et al. 2017).

More pretraining data consistently increases accuracy, especially for larger models. It has no consistent effect on calibration (Figure 3). In particular, after temperature scaling, ECE is essentially unchanged across this 300-fold increase in pretraining dataset size (e.g. for BiT-R50x1 pretrained on IMAGENET, IMAGENET-21K and JFT-300, the ECEs are 0.0185, 0.0182, 0.0185, respectively; for BiT-R101x3, they are 0.0272, 0.0311, 0.0236; Figure 3, bottom). Therefore, regardless of the pretraining dataset, BiT always remains Pareto-dominant over Sim-CLR and Pareto-dominated by ViT and MLP-Mixer in our experiments.

The BiT models compared in Figure 3 differ in both the amount of pretraining data and the duration of pretraining (see Kolesnikov et al. (2020) for details). To further disentangle these variables, we trained BiT models on varying numbers of pretraining examples while holding the number of training steps constant, and vice versa. We find that pretraining dataset size has no significant effect on calibration, while pretraining duration only shifts the model within its accuracy/calibration Pareto set (longer-trained models are more accurate and worse calibrated; Figure 10). These results suggest that pretraining alone cannot explain the differences between model families that we observe.

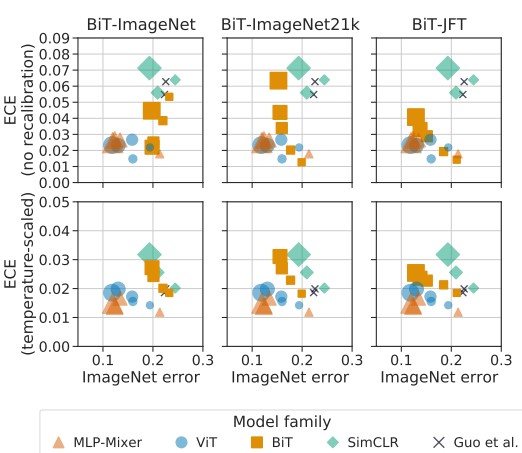

Figure 3: Family differences are not fully explained by the amount of pretraining. Each column shows ECE vs. classification error on ImageNet for BiT models pre-trained with a different dataset: IMAGENET (1.3M images), IMAGENET-21K (12.8M images), or JFT-300 (300M images). The values for other models are provided for reference in light shading (same values as in Figure 2). Note how all BiT models remain in the same relative location between ViT and SimCLR across a 300-fold difference in pretraining data size.

In summary, our results show that some modern neural network families combine high accuracy and state-of-the-art calibration on in-distribution data, both before and after post-hoc recalibration by temperature scaling. In Figure 8 and Appendices E and F, we show that these results generally hold for other measures of model calibration (other ECE variants, Brier score, and model likelihood). Our experiments further suggest that model size and pretraining amount do not fully explain the intrinsic calibration differences between model families. Given that the best-calibrated families (MLP-Mixer and ViT) are non-convolutional, we speculate that model architecture, and in particular its spatial inductive bias, play an important role.

## 4.3 Accuracy and Calibration Under Distribution Shift

For safety-critical applications, the model should produce reasonable uncertainty estimates not just in-distribution, but also under distribution shifts that were not anticipated at training time. We first assess out-of-distribution calibration on the IMAGENET-C dataset, which consists of images that have been synthetically corrupted at five different severities. As expected, both classification and calibration error generally increase with distribution shift (Figure 4; Ovadia et al. 2019; Hendrycks & Dietterich 2019). Interestingly, this decay in calibration performance is slower for MLP-Mixer and ViT than for the other model families, both before and after temperature scaling.

Regarding the effect of model size on calibration, we observed some trend towards worse calibration of larger models on in-distribution data. However, the trend is reversed for most model families as we move out of distribution, especially after accounting for confidence bias by temperature scaling (note positive slope of the gray lines at high corruption severities in Figure 4, bottom row). In other words, the calibration of larger models is more robust to distribution shift (Figure 5).

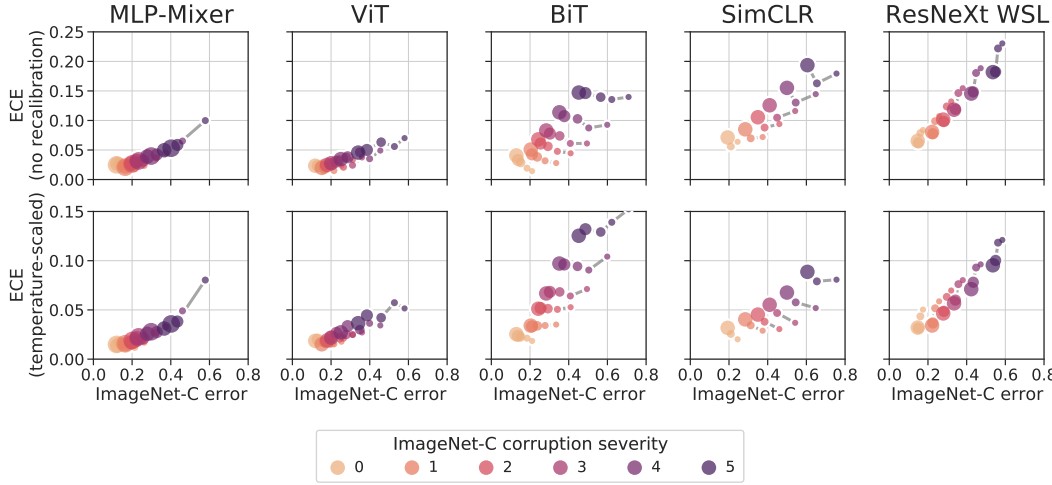

Figure 4: Calibration and accuracy on IMAGENET-C before (top) and after (bottom) temperature scaling on IMAGENET. Severity 0 refers to the clean IMAGENET test set; marker size indicates relative model size within its family (see Table 1 for model details). The calibration of some recent model families, e.g. MLP-Mixer and ViT, is more robust to distribution shift than past models.

We next consider to what degree the insights from in-distribution and IMAGENET-C calibration transfer to natural out-of-distribution data. Previous work on IMAGENET-C suggests that, when comparing recalibration methods, better in-distribution calibration and accuracy do not usually predict better calibration under distribution shift (Ovadia et al., 2019). Here, comparing model families, we find that the performance on several natural out-of-distribution datasets is largely consistent with that on IMAGENET (Figure 6). In particular, models that are Pareto-optimal (i.e. no other model is both more accurate and better calibrated) on IMAGENET remain Pareto-optimal on the OOD datasets. Further, we observe a strong correlation between accuracy and calibration on the OOD datasets. This relationship is consistent across models within a family and across datasets, over a wide range of accuracies (Figure 11).

These results suggest that larger and more accurate models, and in particular MLP-Mixer and ViT, can maintain their good in-distribution calibration even under severe distribution shifts. Based on the observed relationship between calibration and accuracy, we can reasonably hope that good calibration on in-distribution data (and anticipated distribution shifts) generally translates into good calibration on unanticipated out-of-distribution data, similar to what has been observed for accuracy (Djolonga et al., 2020).

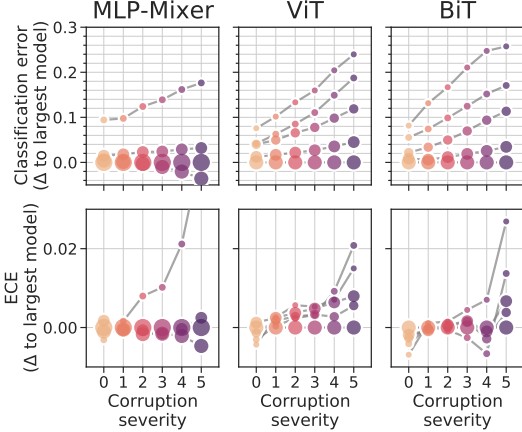

### 4.4 Relating Accuracy and Calibration Within Model Families

Our data suggest that most model families lie on different Pareto sets in the accuracy/calibration space, which establishes a clear preference *between* families. We next consider how to compare individual models *within* a family (or more specifically, within a Pareto set), where one model is more accurate but worse calibrated, and the other is less accurate but better calibrated. Which model should a practitioner choose for a safety-critical application?

Figure 5: Classification error and ECE for the top three families on IMAGENET-C, relative to the largest model variant in each family. As distribution shift increases, both errors tend to increase more slowly for larger models. Also note that changes in ECE are much smaller than changes in classification error.

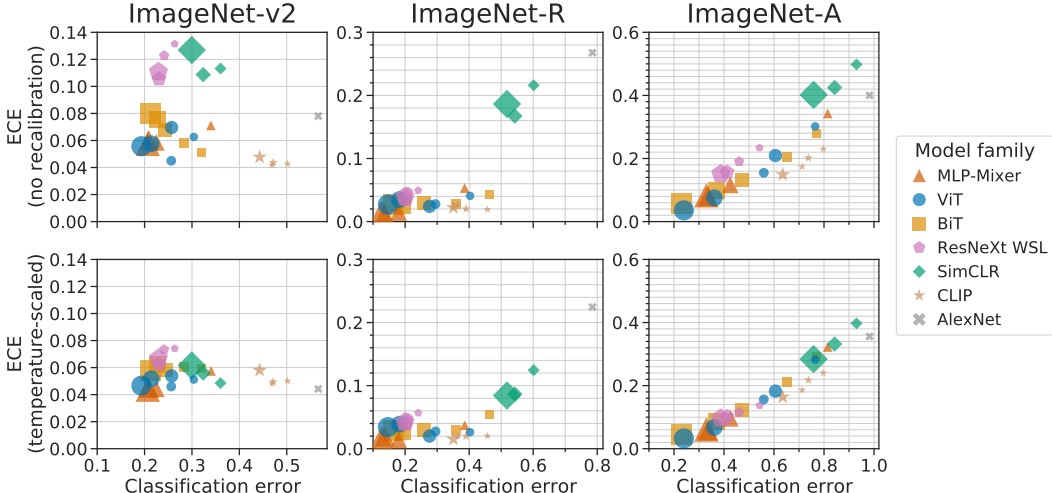

Figure 6: Calibration and accuracy before (top row) and after (bottom row) temperature scaling on out-of-distribution benchmarks. Marker size indicates relative model size within its family. IMAGENET-R and IMAGENET-A use a reduced subset of 200 classes; we follow the literature and select the subset of the model logits for these classes before evaluation. Out-of-distribution calibration tends to correlate with in-distribution calibration (Figure 1) and out-of-distribution accuracy.

The answer depends on the cost structure of the specific application (Hernández-Orallo et al., 2012). As an example, consider the scenario of *selective prediction*, which is common in medical diagnosis. In this task, one can choose to ignore the model prediction ("abstain") at a fixed cost if the prediction confidence is low, rather than risking a (more costly) prediction error.

Figure 7 compares the expected cost for two BiT variants, one with better classification error (R152x4, by 0.08), and one with better ECE (R50x1, by 0.009). For abstention rates up to 70% (which covers most practical scenarios with abstention rates low enough for the model to be useful), the model with better accuracy has a lower overall cost than the model with better ECE. The same is true for all other model families we study (Appendix B.3). For these families and this cost scenario, a practitioner should therefore always choose the most accu-

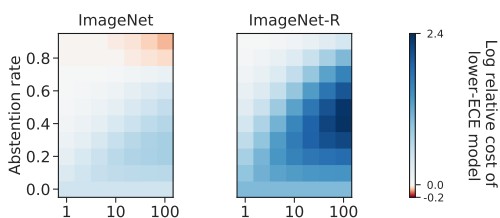

Figure 7: Relative cost of BiT-R152x4 and BiT-R50x1 models in a selective prediction scenario, computed as a combination of the misclassification and abstention costs at a given cost ratio (x-axis) and abstention rate (y-axis). Blue indicates regions where the higher-accuracy model (R152x4) achieves a lower cost than the better-calibrated model (R50x1). The accuracy advantage outweighs the calibration advantage for practical rejection rates, across all tested abstention costs.

rate available model regardless of differences in calibration. Ultimately, real-world cost structures are complex and may yield different results; Figure 7 presents one common scenario with downstream ramifications for the importance of the calibration differences compared to accuracy.

## 5   Pitfalls and Limitations

For this study, we approached calibration with a simple, practical question: *Given two models, one more accurate and the other better calibrated, which should a practitioner choose?* While working towards answering this question, we encountered several pitfalls that complicate the interpretation of calibration results.

Measuring calibration is challenging, and while the quantity we want to estimate is well specified, the estimator itself can be biased. There are two sources of bias: (i) from estimating ECE by binning, and (ii) from the finite sample size used to estimate the per-bin statistics.

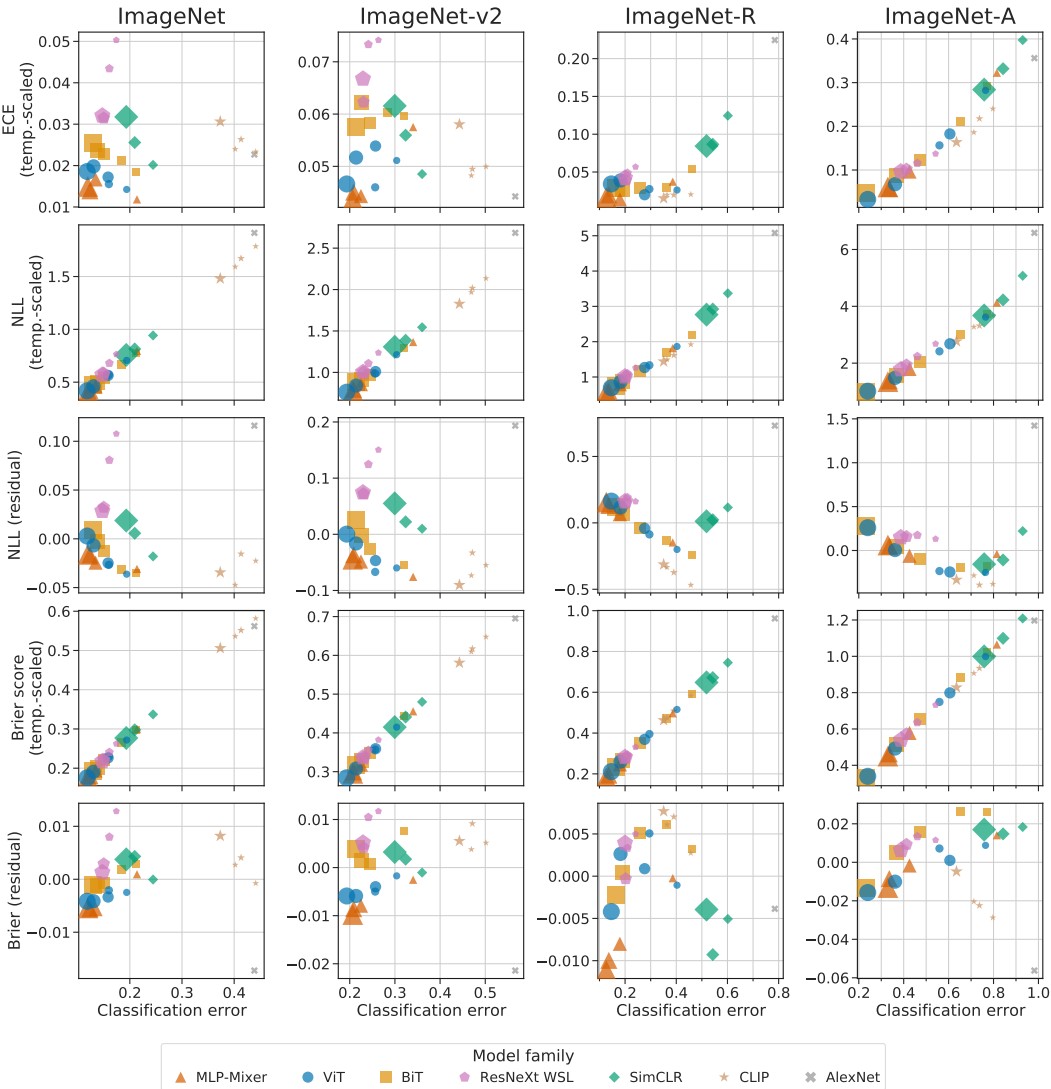

Figure 8: Alternative calibration metrics: negative log-likelihood (NLL) and Brier score. For comparison, the first row shows ECE as in Figure 6. Since NLL, Brier score, and classification error are all highly correlated, we also provide the residuals of NLL and Brier score after regressing out classification error (third and fifth row). Specifically, we first fit a linear regression $y_i = \beta_0 + \beta_1 x_i$, where $x_i$ is the classification error and $y_i$ is the calibration measure of model $i$. We then report the residual $y_i - (\beta_0 + \beta_1 x_i)$ on the $y$-axis of the plots in the third and fifth row. The residuals show which models have better (or worse) NLL and Brier score than what can be expected from their accuracy alone. The relationships between model families are largely similar across all calibration metrics.

The first of these biases is always negative (Kumar et al., 2019), while the second one is always positive. Thus, the estimator can both under- and over-estimate the true value, and the magnitude of the bias can depend on multiple factors. In practice, this means that the ranking of models depends on which ECE variant is chosen to estimate calibration (Nixon et al., 2019). As we show below, this is especially problematic for the positive bias, because this bias *depends on the accuracy of the model*. It is therefore possible to arrive at opposite conclusions about the relationship between accuracy and calibration, depending on the chosen bin size (Figure 9), especially when comparing models with widely varying accuracies.

Intuitively, a larger number of bins implies fewer points per bin and thus higher variance of the estimate of the model accuracy in each bin, which adds positive bias to the estimate of ECE. More formally, to estimate accuracy$(B_i)$ precisely, we need a number of samples inversely proportional to the standard deviation $\sqrt{p_i(1-p_i)}/|B_i|$, where $p_i$ is the expected accuracy in $B_i$. This indicates that the bias would be smaller for models with extreme average accuracies (i.e. close to 0 or 1) and larger for models with an accuracy close to 0.5. A detailed analysis reveals additional effects that further reduce the bias for higher-accuracy models (Appendix C). In particular, if we estimate $(\mathbb{E}[\text{accuracy}(B_i)] - \mathbb{E}[\text{confidence}(B_i)])^2$ for any bin $i$ with $n_i$ samples (using the sample means of the confidences and the accuracies), the bias can be shown to be equal to (conditioning on $X \in B_i$ omitted for brevity)

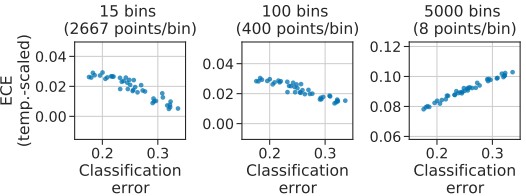

Figure 9: The effect of binning-induced bias in ECE depends on accuracy. Each dot represents a BiT ResNet model. Plotted models differ in model size, pretraining dataset size, and pretraining duration. All models are fine-tuned and evaluated on IMAGENET. After temperature scaling, there is a near-linear relationship between ECE and classification error. However, whether this relationship is positive or negative depends on the number of bins used for estimating ECE. This effect is explained by an accuracy-dependent bias that increases with the number of bins.

$$\frac{1}{n_i}\big(\mathbb{V}[A] + \mathbb{V}[C] - 2\text{Cov}[C, A]\big), \tag{5}$$

where $A = [\![Y \in \arg\max f(X)]\!]$ and $C = \max f(X)$. Hence, from Equation 5 we can conclude that higher accuracy models have a lower bias not only due to higher accuracy (lower $\mathbb{V}[A]$), but also because their outputs correlate more with the correct label (higher covariance).

In addition to a careful choice of bin size, considering accuracy and calibration jointly mitigates this issue, because the Pareto-optimal models rarely change, even if the ranking based on ECE alone does (Appendix E). In Appendices E and F, we provide the main figures of the paper for other ECE variants (number of bins, binning scheme, normalization metric, top-label, all-label, class-wise).

Finally, metrics such as Brier score (Brier, 1950) and likelihood provide alternative assessments of model calibration that do not require estimating expected calibration error. We find that the relationships between model families are consistent across ECE, NLL and Brier score (Figure 8). In particular, the same models (specifically the largest MLP-Mixer and ViT variants) remain Pareto-optimal with respect to the calibration metric and classification error in most cases. The relationship between models is visualized especially clearly after regressing out from the calibration metrics their correlation with classification error (Figure 8, third and fifth row).

## 6 Conclusion

We performed a large study of the calibration of recent state-of-the-art image models and its relationship with accuracy. We find that modern image models are well calibrated across distribution shifts despite being designed with a focus on accuracy. Our results suggest that there is no general trend for recent or highly accurate neural networks to be poorly calibrated compared to older or less accurate models.

Our experiments suggest that simple dimensions such as model size and pretraining amount do not fully account for the performance differences between families, pointing towards architecture as a major determinant of calibration. Of particular note is the finding that MLP-Mixer and Vision Transformers—two recent architectures that are not based on convolutions—are among the best-calibrated models both in-distribution and out-of-distribution. Self-attention (which Vision Transformers employ heavily) has been shown previously to be beneficial for certain kinds of out-of-distribution robustness (Hendrycks et al., 2020a). Our work now hints at calibration benefits of non-convolutional architectures more broadly, for certain kinds of distribution shift. Further work on the influence of architectural inductive biases on calibration and out-of-distribution robustness will be necessary to tell whether these results generalize. If so, they may further hasten the end of the convolutional era in computer vision.

## Acknowledgments and Disclosure of Funding

We thank Carlos Riquelme and Balaji Lakshminarayanan for valuable comments on the manuscript. The authors declare no competing interests.

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
