# Appendix

The Appendix is structured as follows:

# A Models and Datasets

## A.1 Models

Table 1 provides an overview of the models used in this study. Model names link to the used checkpoints, where available.

| Model name | Reference | Variant | Parameters |
| --- | --- | --- | --- |
| AlexNet | Krizhevsky et al. (2012) | 8 layers | 62.4M |
| BiT-L (R50-x1) | Kolesnikov et al. (2020) | ResNet50, 1× width | 25.5M |
| BiT-L (R101-x1) | Kolesnikov et al. (2020) | ResNet101, 1× width | 44.5M |
| BiT-L (R50-x3) | Kolesnikov et al. (2020) | ResNet50, 3× width | 217.3M |
| BiT-L (R101-x3) | Kolesnikov et al. (2020) | ResNet101, 3× width | 387.9M |
| BiT-L (R152-x4) | Kolesnikov et al. (2020) | ResNet154, 4× width | 936.5M |
| CLIP | Radford et al. (2021) | ResNet50-based | 25.5M |
| CLIP | Radford et al. (2021) | ViT-B32-based | 88.3M |
| EfficientNet-NS (B1) | Xie et al. (2020) | 18 layers, 1× width | 7.9M |
| EfficientNet-NS (B3) | Xie et al. (2020) | 31 layers, 1× width | 12.3M |
| EfficientNet-NS (B5) | Xie et al. (2020) | 45 layers, 2× width | 30.6M |
| EfficientNet-NS (B7) | Xie et al. (2020) | 64 layers, 2× width | 66.7M |
| Mixer (B) | Tolstikhin et al. (2021) | B/16, JFT-300m | 59.9M |
| Mixer (L) | Tolstikhin et al. (2021) | L/16, JFT-300m | 280.5M |
| Mixer (H) | Tolstikhin et al. (2021) | H/14, JFT-300m | 589.7M |
| Mixer (H) | Tolstikhin et al. (2021) | H/14, JFT-2.5b | 589.7M |
| ResNeXt-WSL | Mahajan et al. (2018) | ResNeXt 101, 32x8d | 88M |
| ResNeXt-WSL | Mahajan et al. (2018) | ResNeXt 101, 32x16d | 193M |
| ResNeXt-WSL | Mahajan et al. (2018) | ResNeXt 101, 32x32d | 466M |
| ResNeXt-WSL | Mahajan et al. (2018) | ResNeXt 101, 32x48d | 829M |
| SimCLR (1x) | Chen et al. (2020) | ResNet50, 1× width | 25.6M |
| SimCLR (2x) | Chen et al. (2020) | ResNet50, 2× width | 98.1M |
| SimCLR (4x) | Chen et al. (2020) | ResNet50, 4× width | 383.8M |
| ViT (B) | Dosovitskiy et al. (2021) | B/32 | 88.3M |
| ViT (B) | Dosovitskiy et al. (2021) | B/16 | 86.9M |
| ViT (L) | Dosovitskiy et al. (2021) | L/32 | 306.6M |
| ViT (L) | Dosovitskiy et al. (2021) | L/16 | 304.7M |
| ViT (H) | Dosovitskiy et al. (2021) | H/14 | 633.2M |

Table 1: Overview of models used in this study. Per model family, the rows are sorted by increasing marker size in Figure 1 (i.e. approximate relative model size in terms of pretraining compute). We chose a qualitative scale to indicate model size because quantitative measures such as the number of parameters do not always reflect the representational power of a model. For example, ViT-B/16 has slightly fewer parameters than ViT-B/32 but requires more compute and is a more powerful model.

## A.2 Datasets

We evaluate accuracy and calibration the following benchmark datasets:

1. IMAGENET (Deng et al., 2009) refers to the ILSVRC-2012 variant of the ImageNet database, a dataset of images of 1 000 diverse object classes. For evaluation, we use 40 000 images randomly sampled from the public validation set. We reserve the remaining 10 000 images for fitting the temperature scaling parameter.

2. IMAGENETV2 (Recht et al., 2019) is a new IMAGENET test set collected by closely following the original IMAGENET labeling protocol. The dataset contains 10 000 images.

3. IMAGENET-C (Hendrycks & Dietterich, 2019) consists of the images from IMAGENET, modified with synthetic perturbations such as blur, pixelation, and compression artifacts at a range of severities. The dataset includes 15 perturbations at 5 severities each, for a total of 75 datasets. For evaluation, we use the 40 000 images that were not derived from the IMAGENET images we used for temperature scaling.

4. IMAGENET-R (Hendrycks et al., 2020a) contains artificial renditions of IMAGENET classes such as art, cartoons, drawings, sculptures, and others. The dataset has 30 000 images of 200 classes. Following Hendrycks et al., we sub-select the model logits for the 200 classes before computing accuracy and calibration metrics.

5. IMAGENET-A (Hendrycks et al., 2021) contains images that are classified as belonging to IMAGENET classes by humans, but adversarially selected to be hard to classify by a ResNet50 trained on IMAGENET. The dataset has 7 500 samples of 200 classes. As for IMAGENET-R, we sub-select the logits for the 200 classes before computing accuracy and calibration metrics.

In addition, the following datasets are used for pretraining as described in the text:

1. IMAGENET-21K (Deng et al., 2009) refers to the full variant of the ImageNet database. It contains 14.2 million images of 21 000 object classes, organized by the WordNet hierarchy. Each image may have several labels.

2. JFT-300 (Sun et al., 2017) consists of approximately 300 million images, with 1.26 labels per image on average. The labels are organized into a hierarchy of 18 291 classes.

## B   Supplementary Analyses

### B.1   Fine-grained Analysis of Pretraining

Section 4.2 and Figure 3 discuss the effect of the amount of pretraining on accuracy and calibration by comparing models pretrained on three different datasets. Figure 10 provides a more fine-grained analysis. We pretrained BiT models with varying dataset sizes or number of pretraining steps, while holding the other constant. Learning rate schedules were appropriately adapted to the number of steps, i.e. a separate model was trained with a full schedule for each condition, rather than comparing different checkpoints from the same training run. After pretraining, all models were finetuned on IMAGENET as in Kolesnikov et al. (2020).

We find that pretraining dataset size has little consistent effect on calibration error (Figure 10, left). Longer pretraining causes a slight increase in calibration error, but also decreases classification error (Figure 10, right).

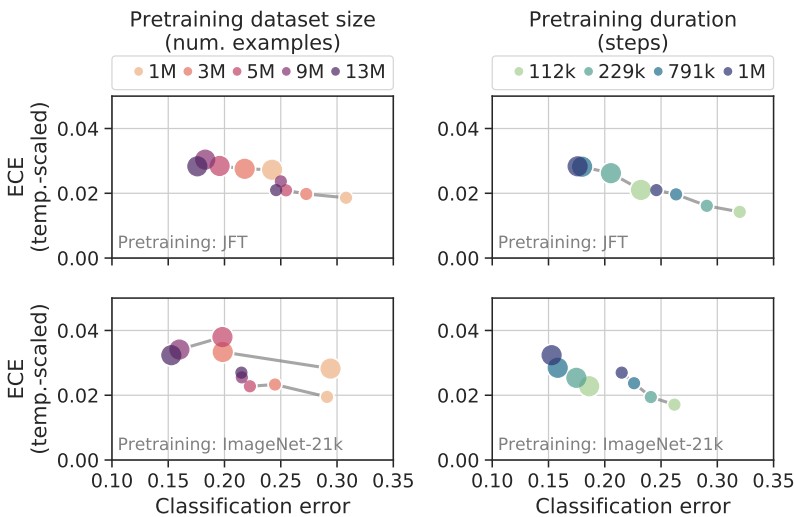

Figure 10: Effect of pretraining dataset size and duration on calibration. Larger dots indicate BiT-R101x3, smaller dots indicate BiT-R50x1. The pretraining datasets are subsampled from JFT-300 (top) or IMAGENET-21K (a larger variant of IMAGENET; bottom). Classification error is on IMAGENET after fine-tuning.

## B.2    Correlation Between Calibration and Accuracy

Figures 4 and 6 show that, across a sufficiently large range of distribution shift, calibration error and classification error are correlated. Figure 11 illustrates this correlation for each model family across model variants and datasets.

In general, it is expected that calibration error and classification error are correlated to some degree due to noise in the model predictions, since adding random noise to the model confidence score would increase both calibration and classification error. Indeed, all model families show a strong positive correlation between calibration and classification error. However, there are consistent differences between model families, reflecting their intrinsic calibration properties. The relationship can be remarkably strong and lawful. For example, a simple power law of the form $y = ax^k$ (where $x$ is classification error and $y$ is ECE) provides a good fit for some model families (e.g. ResNeXt WSL; Figure 11). The parameters of the fit provide a quantitative description of the intrinsic calibration properties of a model family that goes beyond ECE on a specific dataset.

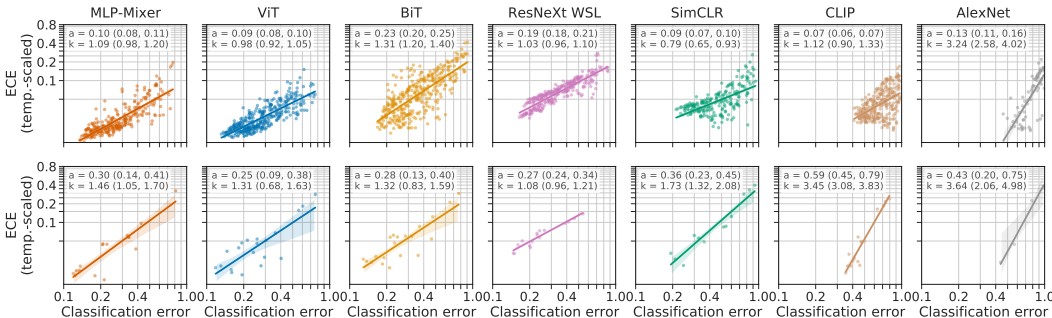

Figure 11: Correlation between ECE and classification error. Each dot represents a different combination of model variant and evaluation dataset (top: IMAGENET-C variants *without* averaging corruptions and severities; bottom: allo other datasets). Lines show power laws of the form $y = ax^k$ where $x$ is classification error and $y$ is ECE. Range in parentheses indicates the 95% confidence interval by bootstrap.

## B.3    Contribution of Accuracy and Calibration to Decision Cost

In Section 4.4, we use a selective prediction task as a practical scenario in which we can quantify the relative impact of accuracy and calibration on the ultimate decision cost incurred by a model user. In this task, the user can either accept a model prediction and incur a misclassification cost if the prediction is wrong, or reject (abstain from) the prediction and incur an abstention cost (which is independent of whether the model prediction would have been correct). This decision is made based on the model's confidence. The total cost therefore depends on both the accuracy and the calibration of the model. A concrete example is a medical diagnosis task in which we can choose to use the model's diagnosis as-is, or refer the case to a human for review. Figure 12 shows cost planes for eight model pairs.

First, we compare models **from the same family** (Figure 12, a–e). In the blue regions, the relative cost for the model with higher accuracy (always the larger model) is lower (better); in red regions, the relative cost of the model with lower accuracy (always the smaller model) is lower (better). For most models and for most practical cost settings, the higher accuracy model is preferred over the better calibrated model. In the ViT family, for example, the bigger model has 0.076 lower classification error (0.194 vs. 0.118) and 0.007 higher ECE (0.017 vs. 0.01). For these models, the cost analysis shows that the difference in classification error outweighs the difference in ECE across all tested misclassification costs and abstention rates.

Next, we compare models **pretrained for a different number of steps** (same models as used in Figure 10) and provide the results in Figure 12, f–g. Again, the models with lower classification error (e.g. for R101x3, 0.176 vs 0.232 in favor of the longer-trained model) reach a lower total cost than the models with lower ECE (e.g. for R101x3, 0.019 vs 0.028 in favor of the shorter-trained model).

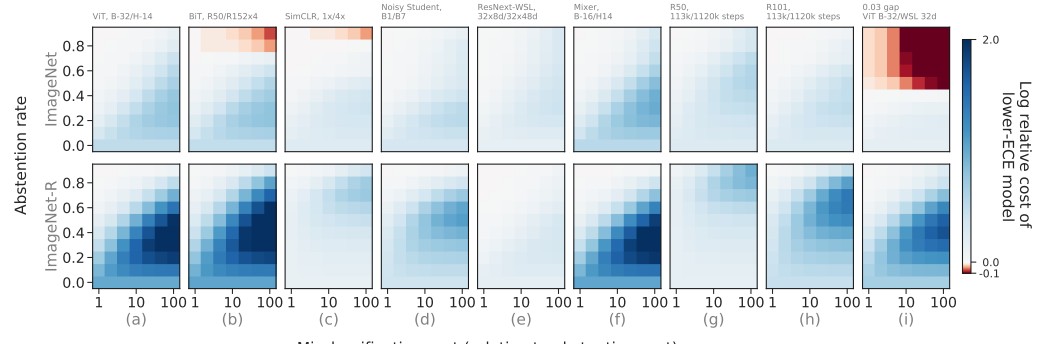

Figure 12: Relative impact of accuracy and calibration in a selective prediction scenario. Each heatmap compares two models and shows the relative cost of the better-calibrated (lower-ECE) model with respect to the other model. Total cost is computed as a linear combination of the misclassification cost and abstention cost at a given cost ratio (x-axis) and abstention rate (y-axis). Compared model pairs are indicated above each column. The top row shows IMAGENET, the bottom row IMAGENET-R. In most scenarios, the higher-accuracy model is preferred over the better-calibrated model (blue regions). Only in a few cases and at very high abstention rates does the difference in calibration outweigh the difference in accuracy (red regions). In other words, for practical abstention rates and across a wide range of abstention costs, the accuracy advantage outweighs the calibration advantage.

Finally, we compare models which attain **similar classification error and ECE difference** (Figure 12, h). In particular, we compare ViT-B/32 and ResNeXt-WSL 32d models. The latter model has 0.03 lower (better) classification error while the ViT model has 0.03 lower (better) ECE. Again, for most practical cost settings, the model with better accuracy has lower cost than (is preferred over) the model with better ECE.

## C  Sampling Bias for $\ell_2$-ECE

In Section 5, we hinted at the fact that the bias of the ECE estimator depends on the model accuracy. Here, we expand on Equation 5 and fully derive the bias for a variant of the ECE score, when we take the squared instead of the absolute differences in each bucket for tractability.

**Lemma 1** *Define the random variables $A = Y \in \arg\max f(X)$ and $C = \max f(X)$, consider the squared ECE metric*

$$ECE_2 = \sum_{i=1}^{m} P(X \in B_i)(accuracy(B_i) - confidence(B_i))^2,$$

*where the $B_i$ represent the $m$ disjoint buckets. If we estimate the per-bin statistics using their sample means, the statistical bias is equal to*

$$\sum_i \frac{1}{n} \mathbb{V}[C - A \mid X \in B_i] = \frac{1}{n} \sum_i (\alpha_i(1 - \alpha_i)(1 - \delta_i) + \mathbb{V}[C \mid X \in B_i]),$$

*where $\alpha_i$ is the accuracy in bucket $B_i$ and $\delta_i$ is the expected difference in the confidences of the correct and incorrect predictions.*

We assume that the buckets are fixed, s.t., there are $n_i$ points in bucket $B_i$, and a total of $n = \sum_i n_i$ points (we will take the expectation over $n_i$). We introduce two random variables — the model confidence by $C = \max f(X)$ and the corresponding true/false indicator by $A = [\![Y \in \arg\max f(X)]\!]$. For each realization $(x_j, y_j)$ we denote by $c_j$ and $a_j$ the corresponding values. We further define for each bucket $B_i$

- $\alpha_i = \mathbb{E}[A \mid X \in B_i]$, the accuracy in bucket $B_i$.
- $\gamma_i = \mathbb{E}[C \mid X \in B_i]$, the expected confidence in bucket $B_i$.

- $\delta_i = \mathbb{E}[C \mid A = 1, X \in B_i] - \mathbb{E}[C \mid A = 0, X \in B_i]$, the confidence difference for the correct and wrong predictions.
- $\bar{c}_i = \sum_{c \in B_i} c/n_i$, the sample average confidence in bucket $B_i$.
- $\bar{a}_i = \sum_{a \in B_i} a/n_i$, the sample average accuracy in bucket $B_i$.

We consider the squared ECE $\ell_2$ loss, which after bucketing is equal to

$$S^2 = \sum_i P(X \in B_i)(\underbrace{\mathbb{E}[C \mid X \in B_i]}_{\gamma_i} - \underbrace{\mathbb{E}[A \mid X \in B_i]}_{\alpha_i})^2, \text{ and the corresponding sample estimate is}$$

$$\hat{S}^2 = \sum_i \frac{n_i}{n}(\frac{1}{n_i}\sum_{i \in B_i} c_i - \frac{1}{n_i}\sum_{j \in B_i} a_j)^2.$$

The goal is to understand the bias $\hat{S}^2 - S^2$. Note that

$$\mathbb{E}[(\bar{c}_i - \bar{a}_i)^2 \mid n_i] = (\gamma_i - \alpha_i)^2 + \mathbb{V}[\bar{c}_i - \bar{a}_i \mid n_i] = (\gamma_i - \alpha_i)^2 + \frac{1}{n_i}\mathbb{V}[C - A \mid X \in B_i].$$

We further have

$$\mathbb{V}[C - A \mid X \in B_i] = \mathbb{V}[C \mid X \in B_i] + \mathbb{V}[A \mid X \in B_i] - 2\text{Cov}[C, A \mid X \in B_i].$$

Hence, we have that

$$\mathbb{E}[\hat{S}^2] = \mathbb{E}[\sum_i \frac{n_i}{n}(\frac{1}{n_i}\sum_{i \in B_i} c_i - \frac{1}{n_i}\sum_{i \in B_i} a_i)^2]$$

$$= \mathbb{E}[\mathbb{E}[\sum_i \frac{n_i}{n}(\frac{1}{n_i}\sum_{i \in B_i} c_i - \frac{1}{n_i}\sum_{i \in B_i} a_i)^2 \mid n_i]]$$

$$= \mathbb{E}[\sum_i \frac{n_i}{n}\left((\gamma_i - \alpha_i)^2 + \frac{1}{n_i}\mathbb{V}[C - A \mid X \in B_i]\right)]$$

$$= \sum_i (\gamma_i - \alpha_i)^2 \underbrace{\mathbb{E}[\frac{n_i}{n}]}_{P(X \in B_i)} + \sum_i \frac{1}{n}\mathbb{V}[C - A \mid X \in B_i]$$

$$= S^2 + \underbrace{\frac{1}{n}\mathbb{V}[C - A \mid X \in B_i]}_{\text{bias}}.$$

We can decompose the covariance as follows (see this MathOverflow answer), using the fact that $A$ is binary:

$$\text{Cov}[C, A \mid X \in G_i] = \mathbb{V}[\alpha_i]\delta_i,$$

Here $\delta_i$ is defined as $\mathbb{E}[C \mid A = 1, X \in B_i] - \mathbb{E}[C \mid A = 0, X \in B_i]$. Now the total bias can be written as

$$\text{bias} = \frac{1}{n}\sum_i \alpha_i(1 - \alpha_i)(1 - 2\delta_i) + \mathbb{V}[C \mid X \in B_i].$$

Note that $\partial_{\alpha_i}\text{bias} = (1 - 2\alpha_i)(1 - 2\delta_i)$, which is negative when $\alpha_j > 1/2$, if we assume we have enough bins so that $\delta_j < 1/2$. Hence, for models that have at least $50\%$ top-1 accuracy, increasing the accuracy reduces the bias.

# D   Model Confidence

An important aspect of the calibration of a model is its average *confidence*, i.e. the systematic bias of the model's scores to be too high (overconfident) or too low (underconfident) compared to the true accuracy. A large fraction of the miscalibration of modern neural networks is typically due to over- or underconfidence (Guo et al., 2017). In this section, we argue that over- and underconfidence are not just a source of miscalibration, but also a confounder that obscures the intrinsic calibration properties of models and makes it harder to compare across model families.

**Quantifying confidence.**   Predictions of overconfident models tend to be overly "peaky" (low entropy), such that an increase in temperature (positive temperature factor) would be necessary to make them optimally confident, and vice versa for underconfident models. We can therefore quantify confidence in terms of the temperature scaling factor by which the logits of the unscaled model would have to be multiplied to provide optimal confidence.

The optimal confidence depends on the model *and* the dataset. Ideally, a model would be optimally confident across all distribution shifts, indicating that its confidence is well calibrated to the difficulty of the data. In practice, most models are slightly overconfident in-distribution, and tend to become more overconfident as data moves further from the training distribution (Figure 13, bottom row).

Models can show the opposite trend if they are *under*confident in-distribution. As an example, we include the EfficientNet-NoisyStudent family in Figure 13. These models tend to be underconfident (optimal temperature factor < 1; Figure 13, bottom right). Underconfident models may paradoxically show *improved* calibration under distribution shift (lower ECE for higher corruption severities), because their underconfidence balances out the general tendency towards overconfidence on OOD data. However, such underconfident models are not better calibrated in general—they are simply biased towards a high level of distribution shift, and are calibrated worse at weak or no distribution shift. A well-calibrated model should have optimal confidence both in- and out-of-distribution.

**Normalizing confidence.**   The example of EfficientNet-NoisyStudent illustrates how confidence bias can confound trends in model calibration. This counfounder can be removed by *temperature scaling* (Guo et al., 2017), i.e. by rescaling model logits to optimize the likelihood on a held-out

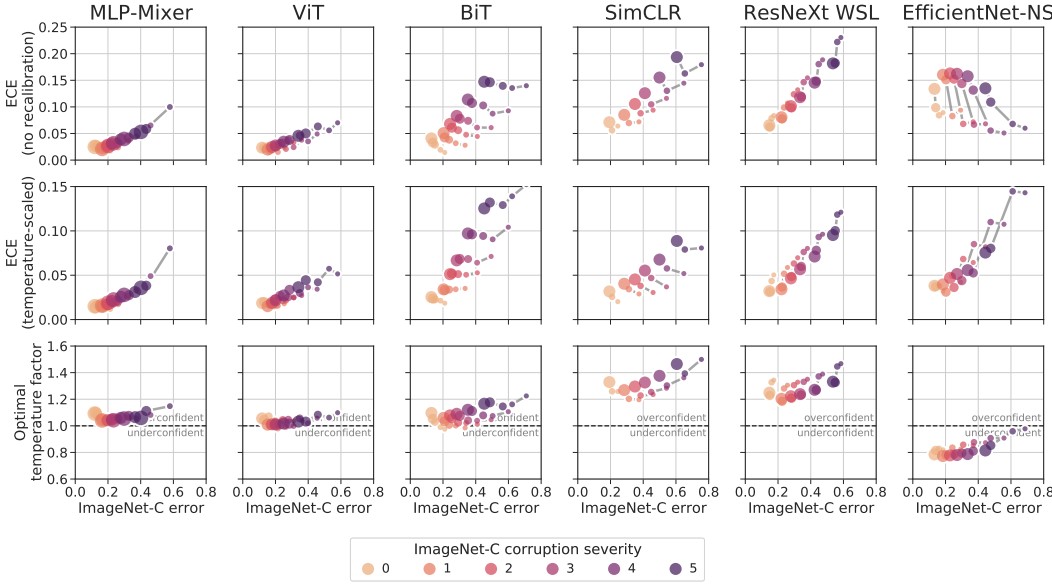

Figure 13: Related to Figure 4. Calibration and accuracy on IMAGENET-C. Here, the model confidence is shown in the third row (top two rows are identical to Figure 4). Model confidence is quantified in terms of the temperature scaling factor by which the logits of the unscaled model would have to be multiplied to provide optimal confidence for a given dataset. Values above 1 mean that the unscaled model is overconfident on the given dataset, and below 1, that the unscaled model is underconfident.

part of the in-distribution dataset (IMAGENET in our case). By removing differences in confidence bias between models, temperature scaling reveals a consistent trend for higher calibration error under distribution shift for all models, including EfficientNet-NoisyStudent (Figure 13, second row). Temperature scaling also reveals consistent differences between model families and trends within families for in-distribution calibration (Figures 2 and 3). We therefore study calibration after temperature scaling, in addition to unscaled calibration error and other calibration metrics (Appendix F), throughout this work. The benefit of temperature scaling for understanding model calibration is separate from its well-established benefit in reducing calibration error (Guo et al., 2017).

**Label smoothing.** One method to directly influence the confidence of a model during training is *label smoothing* (Szegedy et al., 2016). In label smoothing, uniformly distributed probability mass is added to the training targets. This decreases the implied confidence of the targets and thus of the model trained on these targets, which can reduce overfitting and improve accuracy.

Label smoothing has been reported to improve calibration (Müller et al., 2019). Here, we argue that label smoothing creates artificially underconfident models and may therefore improve calibration for a specific amount of distribution shift, but does *not* generally improve the intrinsic calibration properties of a model (i.e. its overall calibration across distribution shifts and datasets).

Figure 14 shows the ECE before and after temperature scaling of models trained with different amounts of label smoothing on IMAGENET and evaluated on IMAGENET-C. Before temperature scaling (Figure 14, left), the best-calibrated models (lowest ECE) are those trained with label smoothing. Depending on the amount of distribution shift (corruption severity), a different amount of label smoothing is necessary to optimize calibration. After temperature scaling on a held-out part of the IMAGENET validation set (Figure 14, center), it becomes clear that training without label smoothing actually results in the lowest ECE across all IMAGENET-C severities. The optimal temperature factor (Figure 14, right) reveals that label smoothing simply biases the model confidence, like temperature scaling, but without targeted optimization. These data suggest that, from a calibration perspective, models should be trained without label smoothing and then recalibrated by temperature scaling *post hoc*.

Label smoothing may explain the anomaly observed for EfficientNet-NoisyStudent under distribution shift (Figure 13, far right). In contrast to all other model families we consider, EfficientNet-NS shows strong *under*confidence before temperature scaling (Appendix D); it is also the only model family trained with label smoothing.

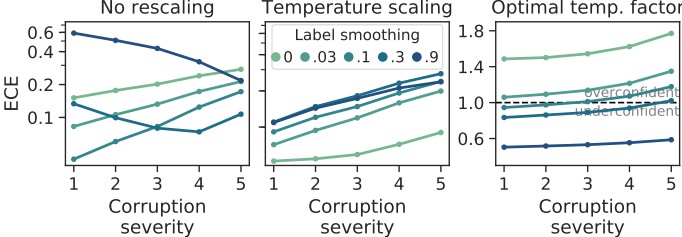

Figure 14: Effect of label smoothing on calibration. EfficientNet-B4 models were trained with the indicated label smoothing on IMAGENET and evaluated on IMAGENET-C. Before rescaling, different amounts of non-zero label smoothing appear to yield the best calibration, depending on distribution shift (left). After temperature scaling, it becomes clear that training without label smoothing is best (center). Label smoothing reduces confidence (right). ResNet architectures show similar behavior.

# E  ECE Variants

As discussed in Sections 2 and 5, while ECE is a well-defined quantity, estimating it requires binning and thus a choice of binning scheme and bin size. In addition, variants of ECE such as root mean squared calibration error (RMSCE, Nixon et al. 2019) exist. In RMSCE, the difference between accuracy and confidence in each bin is $\ell_2$-normalized, in contrast to the $\ell_1$-normalization of standard ECE. This causes larger errors to be upweighted in RMSCE. Further ECE variants consider all classes, instead of just the class with the highest predicted probability (top-label), or consider classes independently and report an average of class-wise ECEs. Different ECE variants may rank models differently (Nixon et al., 2019), which could lead to the conclusion that ECE estimators are fundamentally inconsistent. However, we find that such inconsistencies in model rank are resolved by considering ECE and classification error jointly (Figures 15 to 17). While ranks between models may change across ECE variants, these models differ in classification error, such that it is always clear which model is Pareto-optimal in terms of ECE and classification error. For example, for IMAGENETV2 in Figure 16, the ranking of BiT models (orange squares) changes slightly between some of the ECE variants. However, the models differ so much in classification error that the differences in ECE between metric variants are likely irrelevant (also see Appendix B.3, which shows that differences in classification error typically have a larger influence on decision cost than differences in ECE).

# F  Alternative Calibration Metrics

To confirm that our findings are not dependent on our choice of Expected Calibration Error as our main calibration metric, we provide results for two alternative calibration metrics: negative log-likelihood (NLL) and Brier score (Brier, 1950). Figure 8 in the main text covers IMAGENET, IMAGENETV2, IMAGENET-R, and IMAGENET-A. Results for IMAGENET-C are provided in Figure 18

Furthermore, we provide reliability diagrams (DeGroot & Fienberg, 1983) on IMAGENET for all models, both before (Figure 19) and after (Figure 20) temperature scaling. These diagrams visualize model calibration across the whole confidence range, rather than summarizing calibration into a scalar value.

*Figures for Appendix E and Appendix F are on the following pages.*

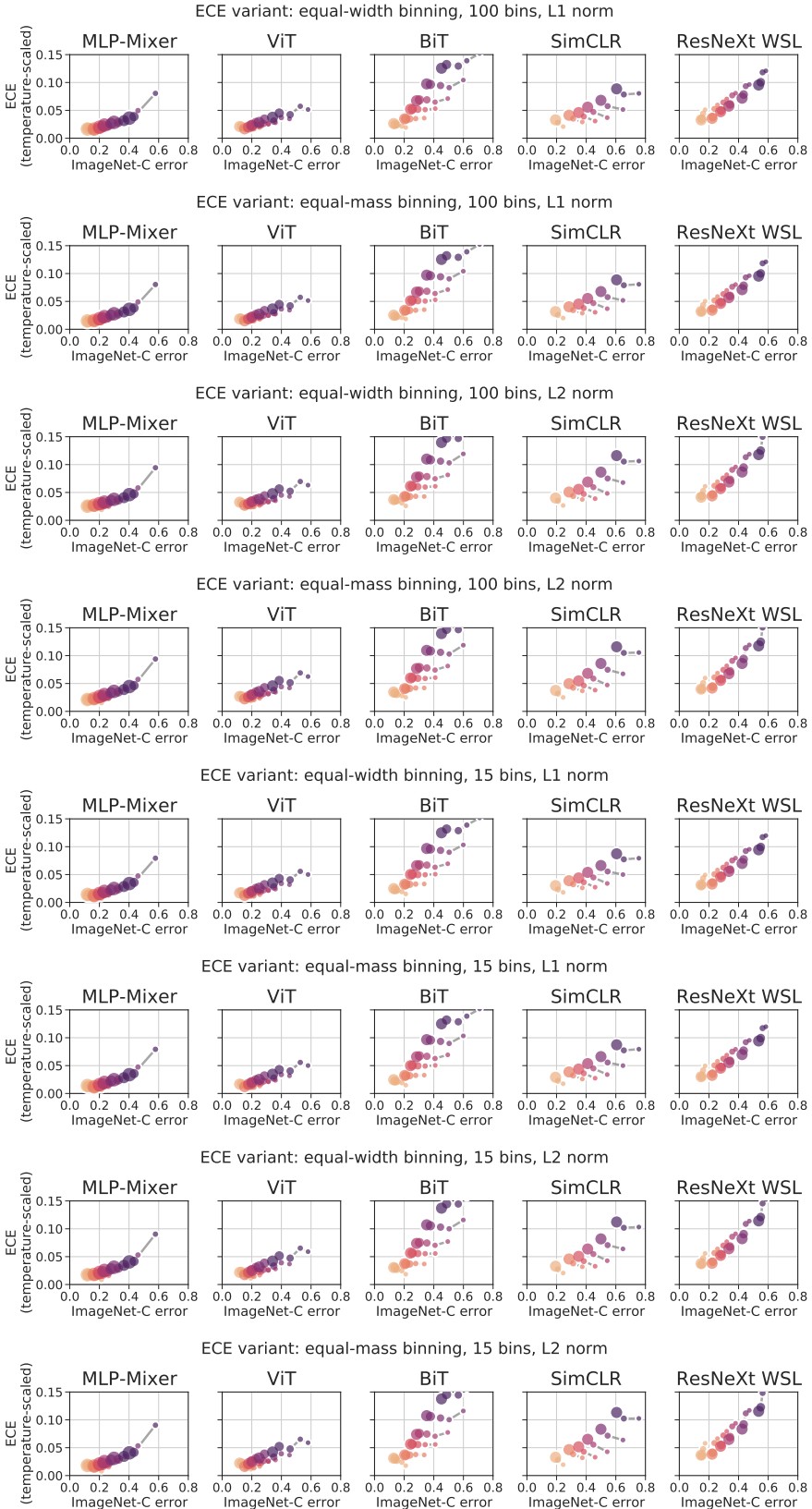

Figure 15: Related to Figure 4. Each row shows the calibration and accuracy on IMAGENET-C as in Figure 4, bottom row (i.e. after temperature scaling), but for different ECE variants. The variant is indicated in the title of each row. While absolute values can differ between variants, relative relationships between models are robust to the metric variant.

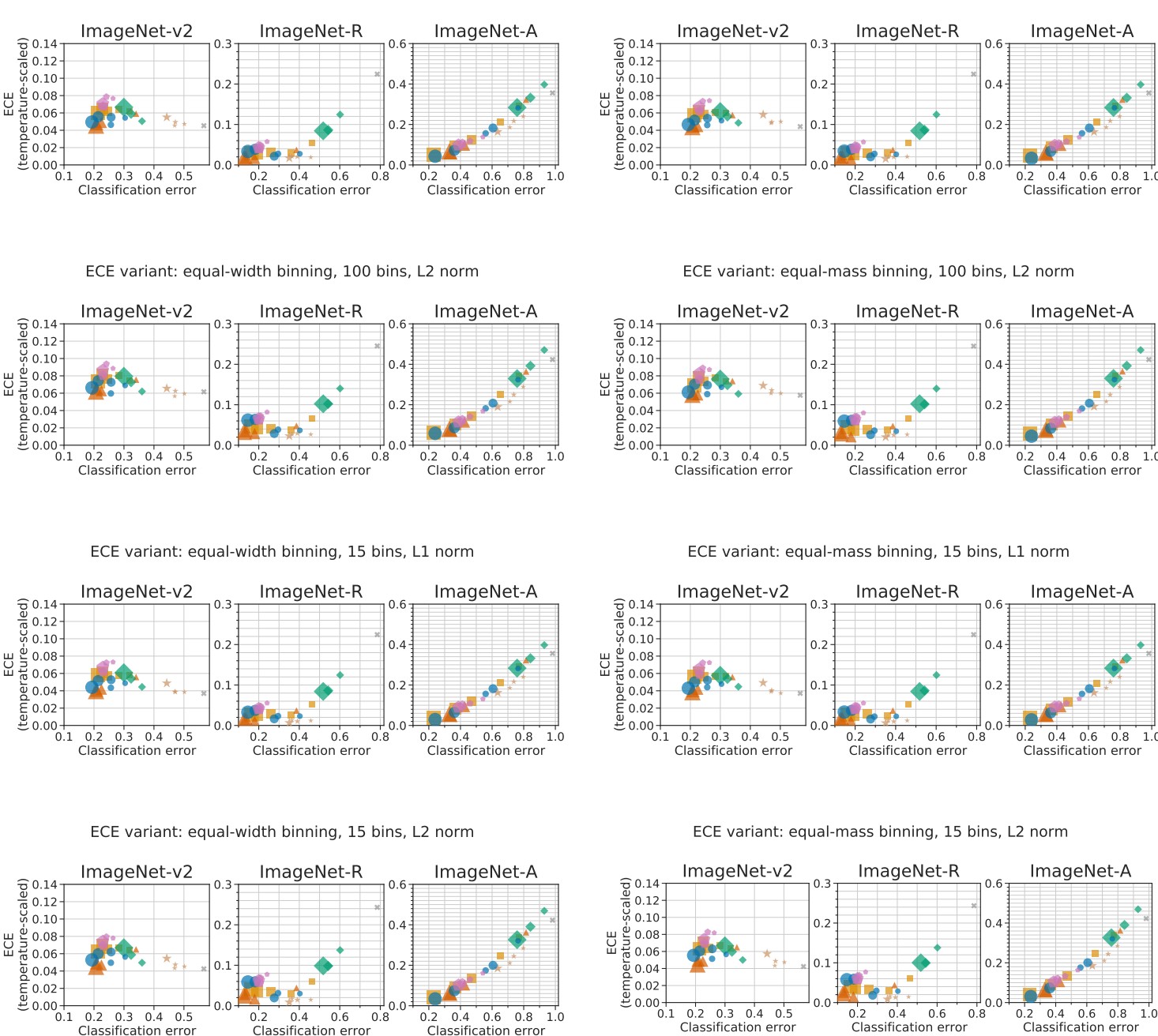

Figure 16: Related to Figure 6. Calibration and accuracy on OOD datasets as in Figure 6, bottom row (i.e. after temperature scaling), but for different ECE variants. The variant is indicated in the title of each set of plots. While absolute values can differ between variants, relative relationships between models are robust to the metric variant.

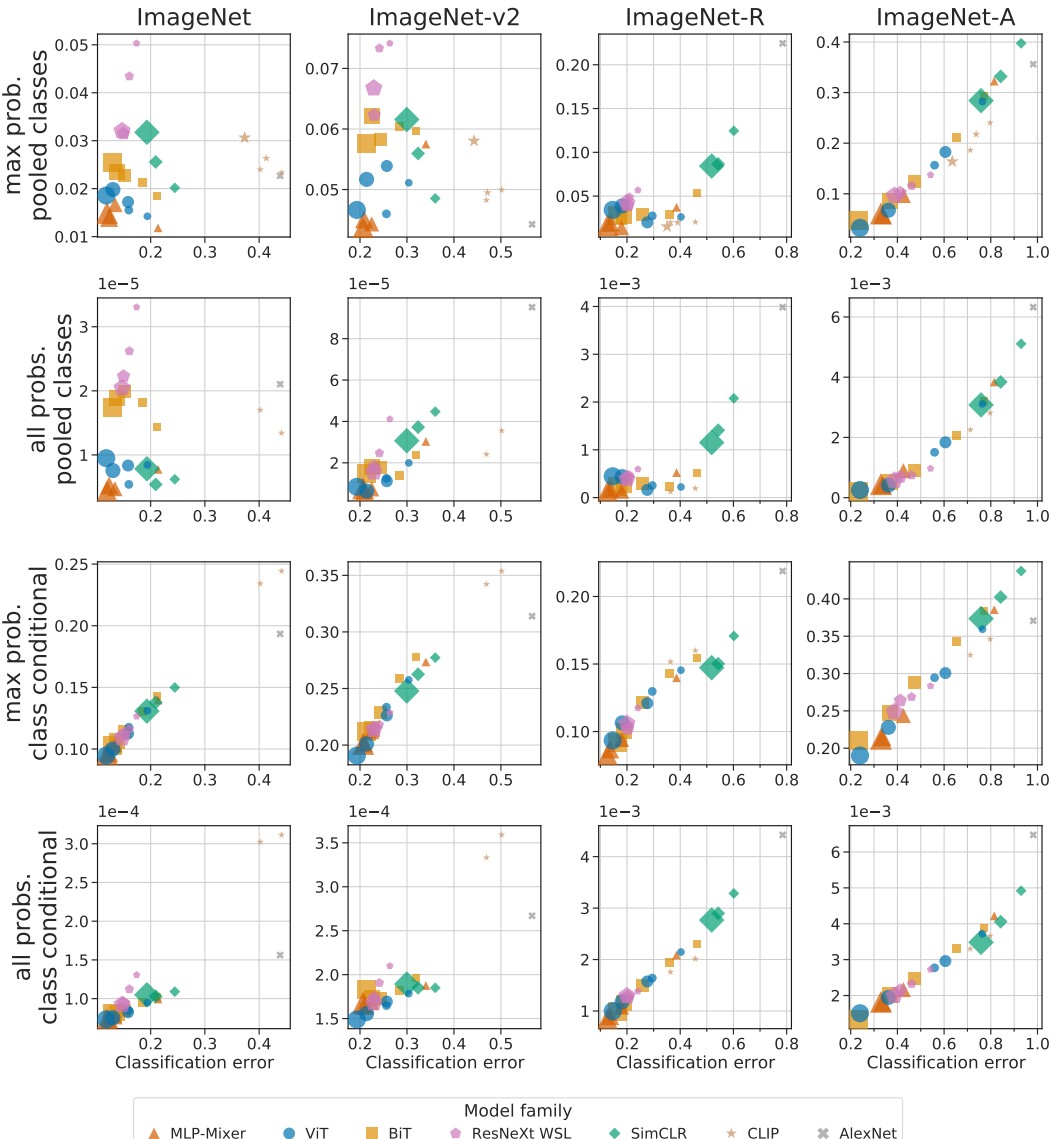

Figure 17: Further ECE variants (after temperature scaling). The top row shows the variant used in the main paper, which considers only the maximum predicted probability ("top-label calibration") and pools across classes. The remaining rows show other variants as discussed in Nixon et al. (2019). L1-normalization and adaptive binning was used in all cases (100 bins for pooled-class metrics; 15 bins for class-conditional metrics). Although the specific rankings between models depend on the ECE variant (Nixon et al., 2019), our main conclusions hold for all variants. Specifically, the same model families tend to be Pareto-optimal across all ECE variants. Also, the relationship between ECE and accuracy is largely consistent across ECE variants.

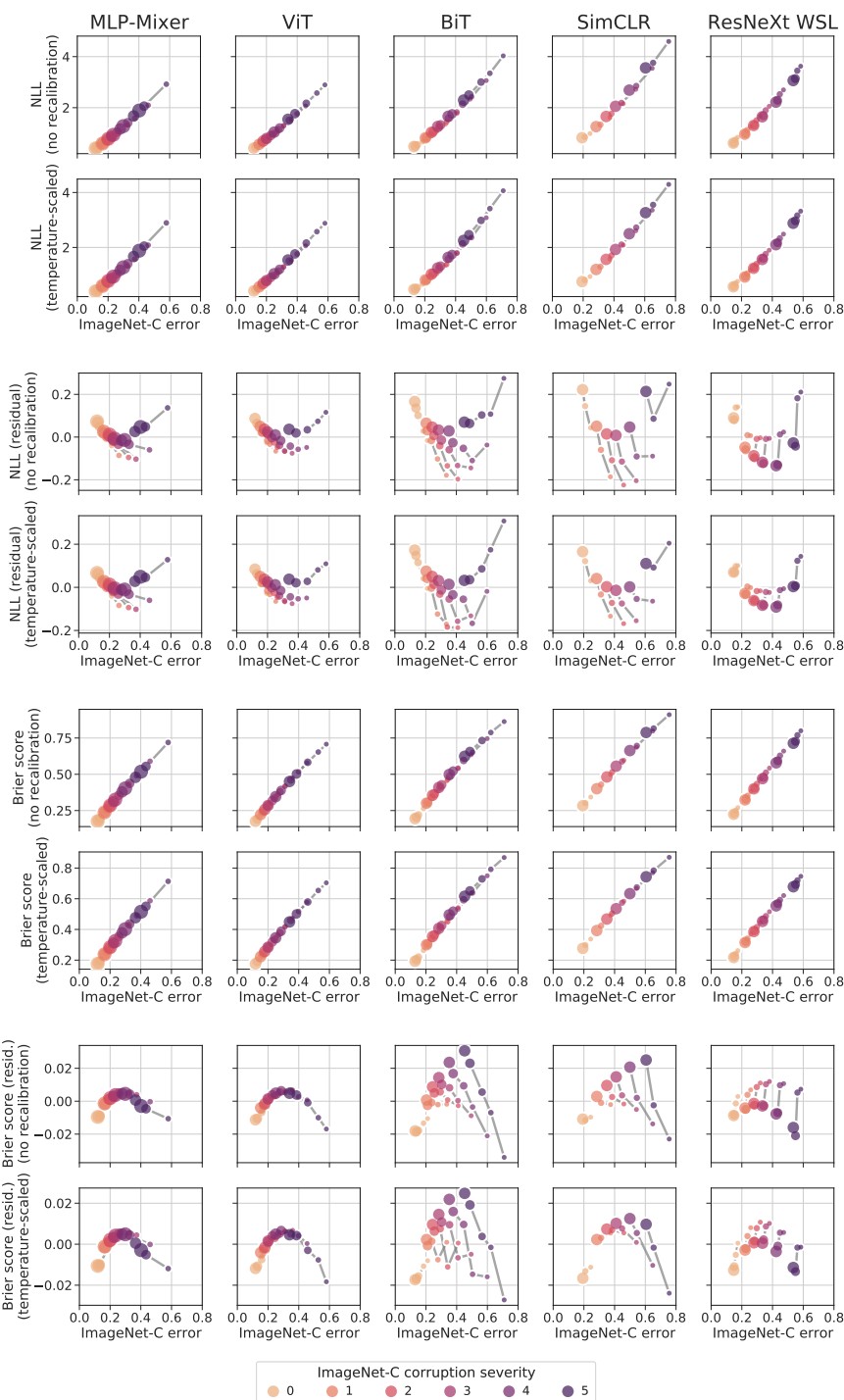

Figure 18: Alternative calibration metrics for IMAGENET-C: negative log-likelihood (NLL) and Brier score. Plotted as in Figure 4. Second and fourth rows show residuals as described in Figure 8.

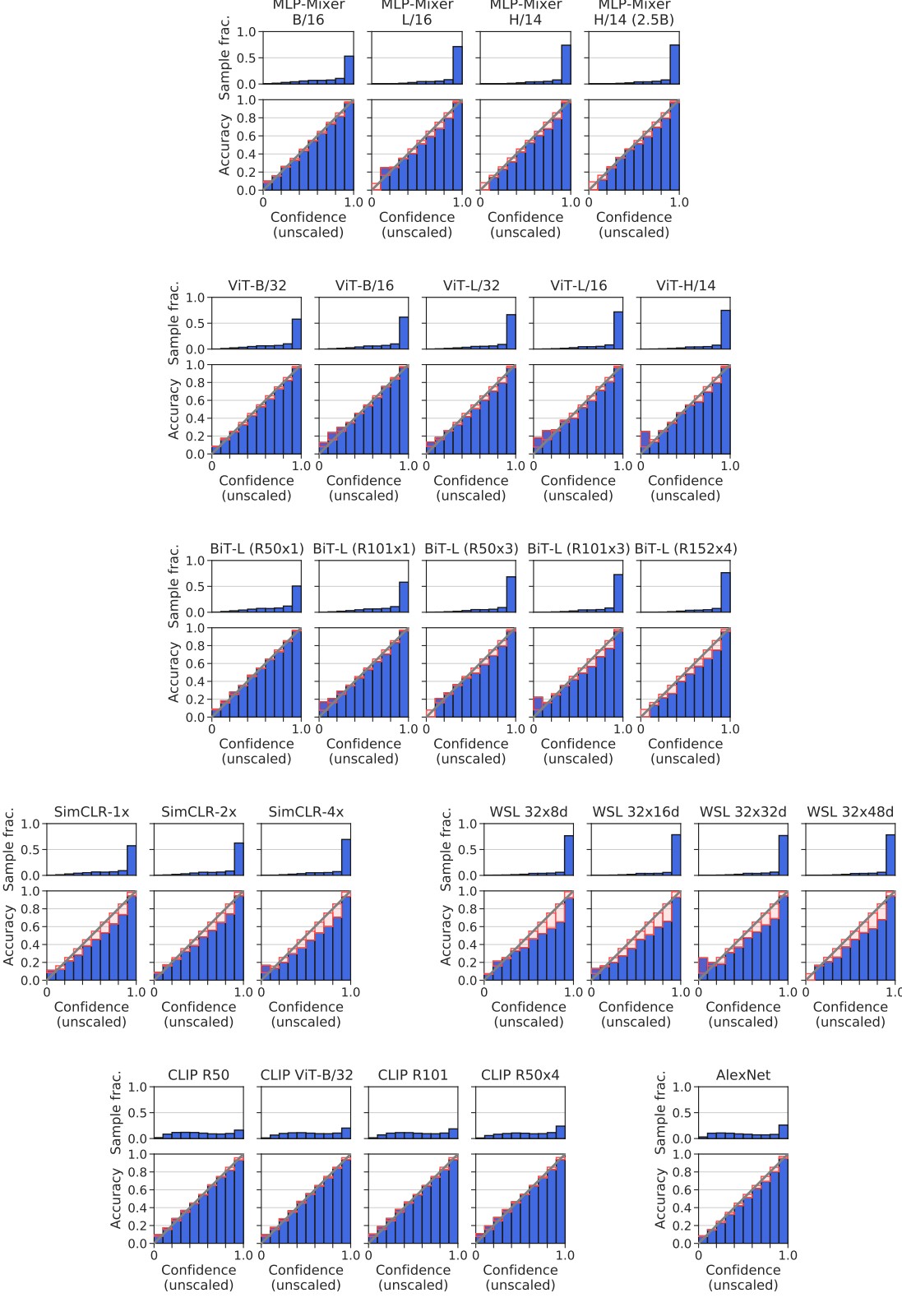

Figure 19: Reliability diagrams on IMAGENET for all models, before temperature scaling. Red boxes indicate the error compared to perfect calibration. The histogram at the top shows the distribution of confidence values for the dataset.

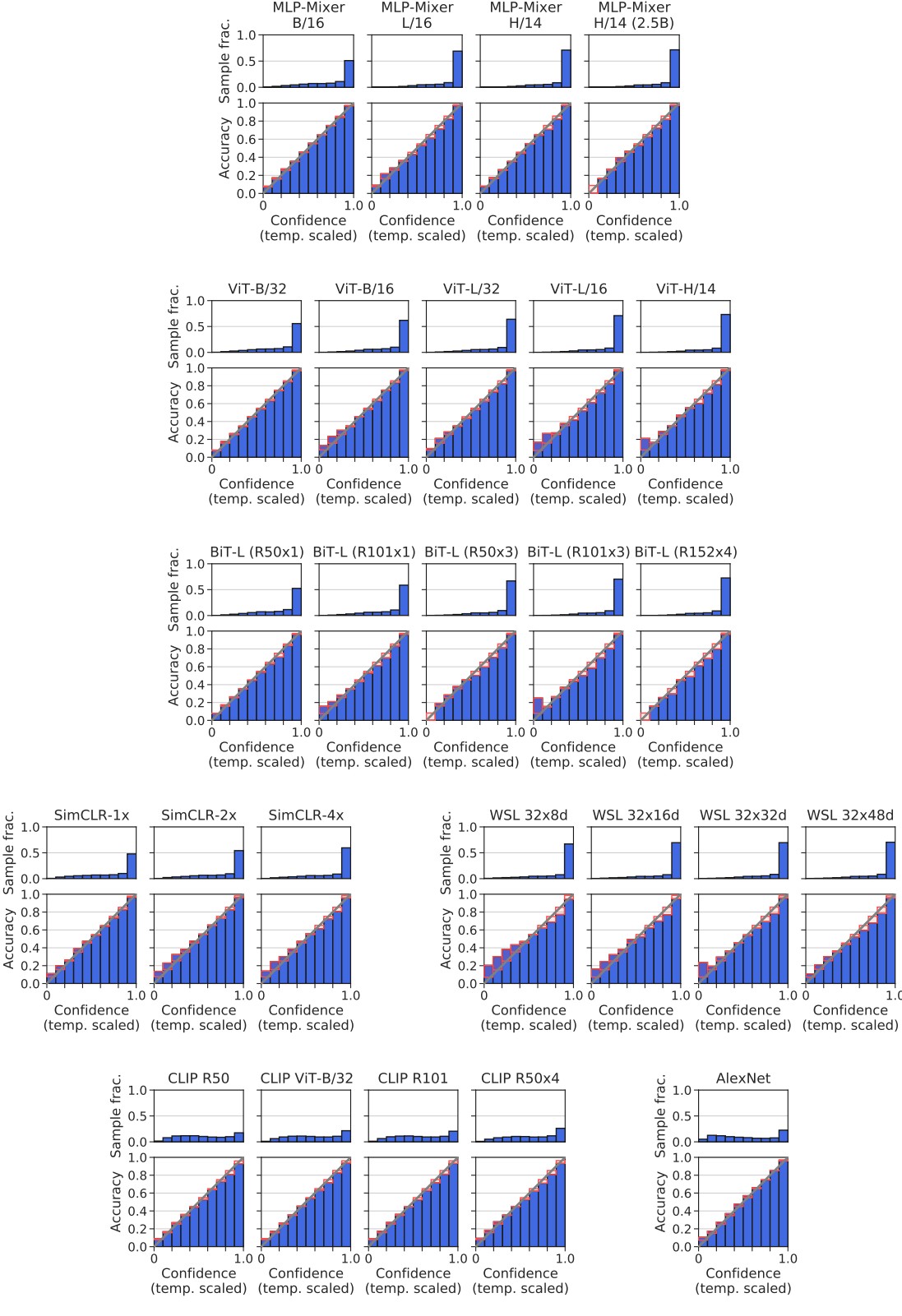

Figure 20: Reliability diagrams on IMAGENET for all models, after temperature scaling. Red boxes indicate the error compared to perfect calibration. The histogram at the top shows the distribution of confidence values for the dataset.