# OpenReview forum: "Revisiting the Calibration of Modern Neural Networks"
_NeurIPS.cc/2021/Conference — NeurIPS 2021 Poster_

### Official Review · Reviewer_9HWK · 2021-07-05

**Rating:** 8
**Confidence:** 4

**Summary:**

The paper evaluates numerous recent models in terms of uncertainty calibration. The models evaluated are recent advances in relation to a previous seminal paper on the topic. The findings of this work indicate that previous trends leading to overconfidence with larger models may not apply to the current state-of-the-art models.

**Limitations And Societal Impact:**

Yes, they discussed the pitfalls of one of the main metrics used in the text. I am not aware of any potential negative societal impacts.

**Main Review:**

# Pros

- Compares many SOTA models which have not been evaluated together in terms of uncertainty thus far.
- Utilizes types of distributionally shifted data which are not considered in most previous calibration works.
- The experiments are thorough, informative, and show that newer families of large non-convolutional models are not overconfident.
- The paper is clearly written and well designed.
- The paper highlights relevant problems with Expected Calibration Error. The authors highlighted the positive bias on ECE of finite samples within bins. To my knowledge, this has not been discussed before.

# Cons

- As there are many different models evaluated, It would be beneficial to see some comparison of the total number of parameters across model families vs. calibration performance of each model.
- Section 5 highlights problems with ECE, which is a crucial axis for all results in the main text and then only refers the reader to the appendix for other measures. The experiments in the appendix (Figure 16) should probably be moved to the main text to provide a measure besides only ECE.
- In the appendix (Figure 16), there is no ImageNet-C dataset. I would like to see the results for NLL and Brier score on this dataset, as it is included in the other results of the main text. The performance at each corruption level may be informative, given that ECE could be biased.
- I am confused by the verbiage in a sentence of Figure 16. How exactly were the 3rd and 5th rows of residual errors calculated?
    - "we also provide the residuals of NLL and Brier score after regressing out
classification error (third and fifth row)"

# Addendum:

My final score derives from the following:

- In terms of originality, the setting is not novel, but the results and conclusions are because the authors seem to have shown that the commonly held belief of DNN overconfidence may not be true anymore for recent models.
- The paper is high quality, very clear and easy to read.
- In terms of significance, I think this paper contributes in both upending the common mantra of overconfidence, and I also think the results here are immediately applicable to practitioners. Some of the models trained here are quite heavy and expensive, so being able to learn from these experiments presents an opportunity to both achieve better calibration and avoid costly and time consuming experimentation.
- The authors have adequately answered any concerns I raised.


**Time Spent Reviewing:**

3-4

---

> ### Author Response · Authors · 2021-08-10
> **Response to review**
>
> Thank you for your thoughtful comments and concrete suggestions. We address them as follows:
>
> 1. **Number of model parameters**: We will add the number of model parameters to Table 1 in the appendix. For context: We chose the purely qualitative scale for model size because quantitative measures such as the number of parameters can be hard to compare across different architectures. For example, one architecture may have more parameters but require less compute, while another has fewer parameters and requires more compute. We will discuss this in the appendix.
> 2. **Alternative calibration metrics in main text**: We will use the additional space in the camera-ready version to move parts of Figure 16 from the appendix to the main paper.
> 3. **Alternative calibration metrics for ImageNet-C**: We have now computed the alternative metrics for ImageNet-C and will add them to the paper (Figure 21 in [anonymous link to new figures](https://storage.googleapis.com/revisiting-calibration/author_response_figures.pdf)).
> 4. **Verbiage of Figure 16 caption**: Thanks for pointing out this issue. We will add the following description to the caption: _"we also provide the residuals of NLL and Brier score after regressing out classification error (third and fifth row)._ ***Specifically, we first fit a linear regression $y_i = \beta_0 + \beta_1 x_i$, where $x_i$ is the classification error and $y_i$ is the calibration measure of model $i$. We then report the residual $y_i - (\beta_0 + \beta_1 x_i)$ on the $y$-axis of the plots in the third and fifth row. The residuals show which models have better (or worse) NLL and Brier score than what can be expected from their accuracy alone.*** "
>
> We hope that we addressed your concerns. Please let us know if you have further questions or comments.

---

> > ### Comment · Reviewer_9HWK · 2021-08-15
> > **Score Update**
> >
> > Thank you for your responses. I have updated my score. I think this paper does a good job of updating prior findings which were true in the past with the current state of art model architectures. I think the results here are immediately applicable to industry and practitioners everywhere who are concerned about calibration, and my score adjustment reflects that importance.

---

### Official Review · Reviewer_Uy9F · 2021-07-14

**Rating:** 6
**Confidence:** 3

**Summary:**

The paper systematically evaluates the relation between model calibration and accuracy. The main finding is that the most recent models, notably those not using convolutions, are among the best calibrated. In addition, the analysis shows that model size and amount of pretraining do not explain these differences. Experimental evaluation is extensive and very carefully performed.

**Main Review:**

The paper focuses on an important and timely question, namely the relation between model architecture, calibration and accuracy. The empirical evaluation performed in the paper is great and very insightful. The hypothesis that convolutions lead to poorly calibrated models is very interesting. Unfortunately, it is not further analysed. Also the paper is mainly empirical, a more theoretical analysis of the problem / interpretation of the results is lacking.

**Time Spent Reviewing:**

2

---

> ### Author Response · Authors · 2021-08-10
> **Response to review**
>
> Thank you for your review and your assessment that our "evaluation is extensive and very carefully performed".
>
> We agree that further analysis of the relationship between calibration and model architecture, especially convolutional vs. non-convolutional architectures, is needed.
>
> While deeper analysis of this question was not feasible given the already extensive scope of our study, we believe that our results will motivate further work in this direction.

---

### Official Review · Reviewer_Yhh5 · 2021-07-16

**Rating:** 7
**Confidence:** 4

**Summary:**

This paper provides a systematic benchmark of recent deep classifiers in terms of the level of calibration.

On the calibration side, this paper focus on the definition of top-label (argmax, confidence) calibration. As a result, the main evaluation metrics are based on the top-label (confidence) ECE. The authors also include accuracy, the proper scoring rules BS and NLL and visualise the reliability diagrams. Temperature scaling is selected as the post-hoc approach.

Various model architectures and training strategies are applied on the imagenet datasets, with additional experiments on out-of-distribution tasks.

Other than using many popular recent approaches, the authors report findings over the relationships on (1) model size and in-distribution calibration, (2) model size and out-of-distribution calibration, (3) accuracy and top-label calibration, (4) model size, training settings and calibration.

**Limitations And Societal Impact:**

Limitation: The authors discuss the limitations of the measures of ECE. So it might be good to include a discussion on calibration tests as mentioned above. It would also be nice to further mention that post-hoc calibration is limited as it depends on a good uncalibrated model. That is, if a model simply outputs the marginal distribution of the target, this isn't much a calibration method can do.

Social impact: I agree with the authors, poorly calibrated models can have a negative impact if applied in society. For instance, to have calibrated probability on things like credit card approval is non-trivial.

**Main Review:**

As suggested by the summary above, this paper is in the position of benchmarking recent deep approaches in terms of calibration. As at the moment there are many ongoing works on the calibration of deep models, it is indeed interesting to have such a paper to provide a systematic comparison. In particular, the reasons for learning uncalibrated results still require further understand, so this paper should provide some empirical results that could be of good impact to other researchers.

That being said, the quality of this paper, therefore, depends on the comprehensiveness of the benchmarks and empirical findings.

On the comprehensiveness side, I am impressed by the selection of deep models and training strategies, and I agree this should bring some nice insights to the deep learning community. Regarding the dataset, while it is mainly designed with the imagenet (hence CV domain), it is understandable given the number of experiments required for such benchmarks.

However, I am less impressed by the choice of calibration definition hence some of the associated metrics. As indicated in the paper, this paper mainly investigates the definition of top-label calibration. While I agree this is the most popular definition since Guo et al 2016, there are indeed some pitfalls regarding that definition. For instance, there will not be any uncalibrated top-label probability smaller than $\frac{1}{K}$ where $K$ is a number of classes. One can easily argue that you should not only focus on the biggest probability values as other small values also represent useful information on the label uncertainty.

This choice of definition further leads to an issue over the empirical findings. One of the contributions claimed by the authors is that the calibration and accuracy are correlated. I tend to think this only holds for the definition of top-label calibration, as accuracy is also a metric about the top-label. For multi-class calibration (df.1) or class-wise calibration, I don't think we can obtain the same results. Therefore, I would be more satisfied if the authors consider explicitly stating they are talking about top-label calibration in both the title and through the main texts. And it would be more than welcome if the authors can include experiments on the aforementioned definitions of calibration (multi-class, class-wise) as well, but it is understandable if the authors prefer not as these experiments are somehow expensive.

On the evaluation metric side, I am happy with the discussion on ECE and including metrics like BS and NLL. Although, there is also a trend of performing statistical tests over the hypothesis of calibration, and report the p-value. It would be nice to see at least some discussion over such methods (https://arxiv.org/abs/1910.11385).

Regarding the post-hoc calibration approaches, while I would like to see more methods, it can easily make the number of experiments grow rather fast(N-models * N-calibration * N-datasets). But one thing I would appreciate is to consider adding (https://proceedings.mlr.press/v80/kumar18a/kumar18a.pdf), which suggests they can improve the level of calibration during the training time.

In general, I think this paper has a good value to be seen by the ML community. I am currently giving a 6 mainly due to the point that the paper doesn't differentiate top-label calibration / general calibration in terms of the correlation on accuracy. I am happy to change the score if upon a discussion with the authors.

=============After author feedback============================

The authors addressed some of my concerns in the response, I am happy to increase the score to 7.

(1) In response to my comments on the selected definition of calibration (hence ECE), the authors add some experiments to further justify their empirical findings.

(2) The authors acknowledge some of the related work and indicate the benchmark should be accessible for others to add additional experiments.

At the moment I am not giving any higher score (e.g. 8) due to the fact that the overall paper still mainly reports empirical findings. But I agree it should provide a good reference for the community.

**Time Spent Reviewing:**

about 4 hours

---

> ### Author Response · Authors · 2021-08-10
> **Response to review**
>
> Thank you for your thoughtful comments and useful suggestions.
>
> ### Additional ECE variants
>
> The main concern raised in the review was the paper's focus on top-label calibration. Following the reviewer's suggestion, **we have now repeated the main analyses using both all-label ECE and class-wise ECE** (Figure 19 in [anonymous link to new figures](https://storage.googleapis.com/revisiting-calibration/author_response_figures.pdf)). Specifically, we provide the following variants:
>
> * **top-label; pooling all classes** (variant used in the original paper)
> * **all-label; pooling all classes** (“not using only max probabilities” in [Nixon et al. 2019](https://arxiv.org/abs/1904.01685))
> * **top-label; class-wise** (“class-conditional” in [Nixon et al. 2019](https://arxiv.org/abs/1904.01685))
> * **all-label; class-wise** (“not using only max probabilities” + “class-conditional” in [Nixon et al. 2019](https://arxiv.org/abs/1904.01685); also known as “marginal calibration error” in [Kumar et al., 2019](https://arxiv.org/abs/1909.10155))
>
> (We use 100 equal-mass bins when pooling classes and 15 equal-mass bins for class-wise metrics, and L1 normalization in all cases.)
>
> Our main conclusions hold for all ECE variants. In particular:
>
> 1. The models that are Pareto-optimal with respect to calibration and accuracy for the top-label/all-class ECE are also Pareto-optimal (or close to optimal) for the other variants.
> 2. The correlation between calibration and accuracy under distribution shift is at least as strong for the alternative variants as for the variant used in the original submission, and even stronger for class-wise ECE.
>
> Some small but interesting differences exist between top-label and all-label ECE. For example, SimCLR is comparatively poorly calibrated when considering only the top label, but is on par with ViT when considering all labels.
>
> The results using all-label ECE suggest that the observed correlation between accuracy and calibration is not just due to a focus on the top label, but a more general relationship. We will add these results to the paper.
>
> ### Statistical tests over the hypothesis of calibration
>
> Thank you for pointing us to [Widmann et al., 2019](https://arxiv.org/abs/1910.11385). As discussed in the paper, we agree that binning-based ECE estimators have limitations, and welcome new estimators with better statistical properties and ways to measure their uncertainty. We will add this method to our discussion of the pitfalls of ECE. In addition, the code and data that we will release with the paper will make it feasible for others to evaluate new metrics implementations in the framework of our paper.
>
> We hope that the additional analyses address your main concerns. Please let us know if you have further questions or comments.

---

### Decision · Program_Chairs · 2021-09-27

**Decision:**

Accept (Poster)

**Comment:**

This paper studies the question of calibration in neural network image classifiers. Prior and widely cited empirical work in this area showed that state-of-the art models (at the time) could exhibit quite poor calibration properties. This paper empirically demonstrates that these same issues appear to be much less of a concern with recent, high-performing deep architectures including the MLP-Mixer and Vision Transformers in both the in-distribution setting and the out-of-distribution setting. The primary novelty in this work comes from re-visiting the question of calibration in light of recent advances in model architectures. The empirical results have strong practical significance as they indicate that past findings about calibration in convolutional models appear not to apply to recent deep architectures. The experiments conducted and metrics chosen are sound. Following the author response and discussion, the consensus is that the paper should be accepted. In revising the manuscript, the authors should include the additional results described during the discussion and take the reviewers' suggestions on the partitioning of material between the main body of the paper and the supplemental material into account.